# Feedback-guided Data Synthesis for Imbalanced Classification

**Reyhane Askari-Hemmat**[1,2,3,†]   **Mohammad Pezeshki**[1]*   **Florian Bordes**[1,2,3] *
**Michal Drozdzal**[1]   **Adriana Romero-Soriano**[1,2,4]

[1]**FAIR at Meta** [2]**Mila** [3]**Université de Montréal** [4] **McGill University, Canada CIFAR AI chair**

**Reviewed on OpenReview:** https://openreview.net/forum?id=IHJ5OohGwr

## Abstract

Current status quo in machine learning is to use static datasets of real images for training, which often come from long-tailed distributions. With the recent advances in generative models, researchers have started augmenting these static datasets with synthetic data, reporting moderate performance improvements on classification tasks. We hypothesize that these performance gains are limited by the lack of feedback from the classifier to the generative model, which would promote the *usefulness* of the generated samples to improve the classifier's performance. In this work, we introduce a framework for augmenting static datasets with useful synthetic samples, which leverages one-shot feedback from the classifier to drive the sampling of the generative model. In order for the framework to be effective, we find that the samples must be *close to the support* of the real data of the task at hand, and be sufficiently *diverse*. We validate three feedback criteria on a long-tailed dataset (ImageNet-LT, Places-LT) as well as a group-imbalanced dataset (NICO++). On ImageNet-LT, we achieve state-of-the-art results, with over 4% improvement on underrepresented classes while being twice efficient in terms of the number of generated synthetic samples. Similarly, on Places-LT we achieve state-of-the-art results as well as nearly 4% improvement on underrepresented classes. NICO++ also enjoys marked boosts of over 5% in worst group accuracy. With these results, our framework paves the path towards effectively leveraging state-of-the-art text-to-image models as data sources that can be queried to improve downstream applications [1].

## 1 Introduction

In the recent year, we have witnessed unprecedented progress in image generative models (Ho et al., 2020; Nichol et al., 2022; Ramesh et al., 2021; Rombach et al., 2022; Ramesh et al., 2022; Saharia et al., 2022; Balaji et al., 2022; Kang et al., 2023). The photo-realistic results achieved by these models has propelled an arms race towards their widespread use in content creation applications, and as a byproduct, the research community has focused on developing models and techniques to improve image realism (Kang et al., 2023) and conditioning-generation consistency (Hu et al., 2023; Yarom et al., 2023; Xu et al., 2023). Yet, the potential for those models to become sources of data to train machine learning models is still under debate, raising intriguing questions about the *qualities* that the synthetic data must possess to be effective in training downstream representation learning models.

Several recent works have proposed using generative models as either data augmentation or sole source of data to train machine learning models (He et al., 2023; Sariyildiz et al., 2023; Shipard et al., 2023; Bansal & Grover, 2023; Dunlap et al., 2023; Gu et al., 2023; Astolfi et al., 2023; Tian et al., 2023), reporting moderate model performance gains. These works operate in a static scenario, where the models being trained do not provide any *feedback* to the synthetic data collection process that would ensure the *usefulness* of the generated

---

*Equal contribution, † Corresponding author: reyhaneaskari@meta.com.
[1]Code: https://github.com/facebookresearch/Feedback-guided-Data-Synthesis

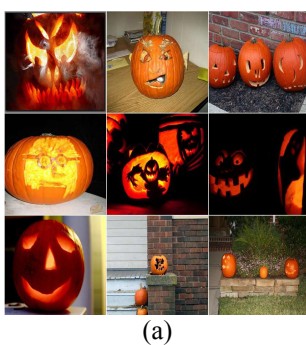 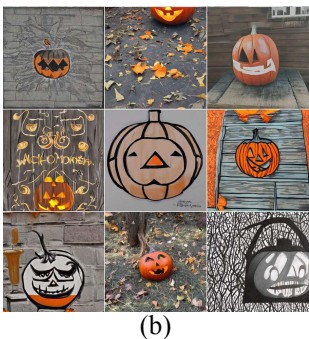 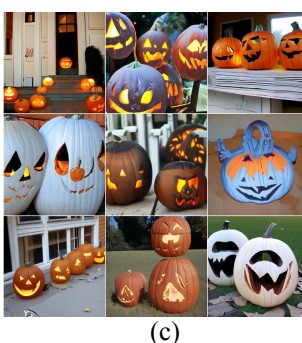

(a)                                    (b)                                    (c)

Figure 1: **Exemplary samples from different distributions.** Subfigures show random samples for *Jack-o-lantern* class coming from: (a) ImageNet-LT; (b) Latent Diffusion Model (LDM-unclip v2-1), conditioned on the text prompt `Jack-o-lantern`; These exhibit a noticeable distribution mismatch with real data. (c) our pipeline, the generated samples more closely align with the real data distribution and present higher diversity, including rare samples such as blue or white pumpkins.

samples. Instead, to achieve performance gains, the proposed approaches often rely on laborious 'prompt engineering' (Gu et al., 2023) to promote synthetic data to be *close to the support of the real data distribution* on which the downstream representation learning model is to be deployed (Shin et al., 2023). Moreover, recent studies have highlighted the limited *conditional diversity* in the samples generated by state-of-the-art image generative models (Hall et al., 2023; Cho et al., 2022b; Luccioni et al., 2023; Bianchi et al., 2022), which may hinder the promise of leveraging synthetic data at scale. From these perspectives, synthetic data still falls short of real data.

Yet, the generative model literature has implicitly encouraged generating synthetic samples that are close to the support of the real data distribution by developing methods to increase the controllability of the generation process (Vendrow et al., 2023). For example, researchers have explored image generative models conditioned on images instead of only text (Casanova et al., 2021; Blattmann et al., 2022; Bordes et al., 2022). These approaches inherently offer more control over the generation process, by providing the models with rich information from a real image without relying on 'prompt engineering' (Wei et al., 2022; Zhang et al., 2022; Lester et al., 2021). Similarly, the generative models literature has aimed to increase sample diversity by devising strategies to encourage models to sample from the tails of their distribution (Sehwag et al., 2022; Um & Ye, 2023). However, the promise of the above-described strategies to improve representation learning is yet to be shown.

In this work, we propose to leverage the recent advances in the generative models to address the shortcomings of synthetic data in representation learning, and introduce feedback from the downstream classifier model to guide the data generation process. In particular, we devise a framework which leverages a pre-trained image generative model to provide *useful*, and *diverse* synthetic samples that *are close to the support of the real data distribution*, to improve on representation learning tasks. Since real world data is most often characterized by long tail and open-ended distributions, we focus on imbalanced classification-scenarios, in which different classes or groups are unequally represented, to demonstrate the effectiveness of our framework. More precisely, we conduct experiments on ImageNet Long-Tailed (ImageNet-LT) (Liu et al., 2019), Places Long-Tailed (Places-LT) and NICO++ (Zhang et al., 2023) and show consistent performance gains *w.r.t.* prior art. Our contributions can be summarized as:

- We devise a diffusion model sampling strategy which leverages feedback from a pretrained classifier in order to generate samples that are useful to improve its own performance.
- We find that for the classifier's feedback to be effective, the synthetic data must lie *close to the support* of the downstream task data distribution, and be sufficiently *diverse*.
- We report state-of-the-art results (1) on ImageNet-LT, with an improvement of 4% on underrepresented classes while using half the amount of synthetic data than the previous state-of-the-art; and (2) we report state-of-the-art results on Places-LT, with an improvement of nearly 4% on underrepresented classes; and (3) on NICO++, with improvements of over 5% in worst group accuracy.

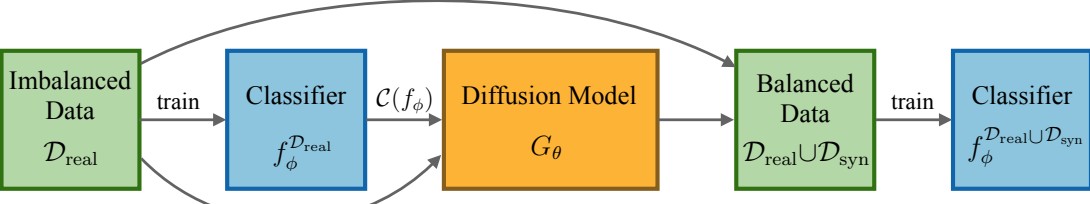

Dual conditioning: A random sample + `text prompt`

Figure 2: **Overview of our framework.** Given an imbalanced dataset, $\mathcal{D}_{\text{real}}$, a classifier $f_\phi(x)$ is initially trained. Knowing that the validation and test sets are balanced, the goal is to create a balanced training set using synthetic data. The Diffusion Model, $G_\theta$, is conditioned on a randomly selected real image and a label-containing text prompt. The model's generation is also guided by feedback, $\mathcal{C}(f_\phi)$, from the classifier to increase usefulness of the synthetic samples. Subsequently, $f_\phi(x)$ is retrained on the combined real and synthetic samples.

Through experiments, we highlight how our proposed approach can be effectively implemented to enhance the utility of synthetic data. See Figure 1 for samples from our framework.

## 2 Background

**Diffusion models.** Diffusion models (Sohl-Dickstein et al., 2015; Song & Ermon, 2019) learn data distributions $p(\boldsymbol{x})$ or $p(\boldsymbol{x}|y)$ by simulating the diffusion process in forward and reverse directions. In particular, Denoising Diffusion Probabilistic Models (DDPM) (Ho et al., 2020) add noise to data points in the forward process and remove it in the reverse process. The continuous-time reverse process in DDPM is given by, $d\mathbf{x}_t = \left[\mathbf{f}(\mathbf{x}_t, t) - g^2(t)\nabla \log p^t(\mathbf{x}_t)\right] dt + g(t)d\mathbf{w}_t$, where $t$ indexes time, and $\mathbf{f}(\mathbf{x}_t, t)$ and $g(t)$ are drift and volatility coefficients. A neural network $\epsilon_\theta^{(t)}(\boldsymbol{x}_t)$ is trained to predict noise in DDPM, aligning with the score function $\nabla \log p^t(\mathbf{x}_t)$. Given a trained model $\epsilon_\theta^{(t)}(\boldsymbol{x}_t)$, Denoising Diffusion Implicit Models (DDIM) (Song et al., 2020), a more generic form of diffusion models, can generate an image $\boldsymbol{x}_0$ from pure noise $\boldsymbol{x}_T$ by repeatedly removing noise, getting $\boldsymbol{x}_{t-1}$ given $\boldsymbol{x}_t$ (Song et al., 2020):

$$\boldsymbol{x}_{t-1} = \sqrt{\alpha_{t-1}} \underbrace{\left( \frac{\boldsymbol{x}_t - \sqrt{1-\alpha_t}\epsilon_\theta^{(t)}(\boldsymbol{x}_t)}{\sqrt{\alpha_t}} \right)}_{\text{`` predicted } \boldsymbol{x}_0 \text{ ''}} + \underbrace{\sqrt{1-\alpha_{t-1}-\sigma_t^2} \cdot \epsilon_\theta^{(t)}(\boldsymbol{x}_t)}_{\text{``direction pointing to } \boldsymbol{x}_t \text{ ''}} + \underbrace{\sigma_t\epsilon_t}_{\text{random noise}}, \tag{1}$$

with $\alpha_t$ and $\sigma_t$ as time-dependent coefficients and $\epsilon_t \sim \mathcal{N}(\boldsymbol{0}, \boldsymbol{I})$ being standard Gaussian noise.

**Classifier-guidance in diffusion models.** Guidance in latent diffusion models involves leveraging additional information, such as class labels or text prompts, to condition the generated samples on. This modifies the score function as follows:

$$\nabla_x \log p_\gamma(x|y) = \nabla_x \log p(x) + \gamma \nabla_x \log p(y|x), \tag{2}$$

where $\gamma$ is a scaling factor. The term $\nabla_x \log p(y|x)$ is generally modeled either as classifier-guidance (Dhariwal & Nichol, 2021) or classifier-free guidance (Ho & Salimans, 2022). In the Latent Diffusion Model (LDM) used in this paper, $\nabla_x \log p(y|x)$ is modeled in a classifier-free approach and $\gamma$ controls its guidance strength.

## 3 Methodology

Figure 2 presents an overview of our proposed approach. We assume access to a pre-tained diffusion model, which takes as input an image and a text prompt, and produces an image consistent with the inputs. We train a classifier $f_\phi$ on an imbalanced dataset of real images, $\mathcal{D}_{\text{real}}$. This initial classifier serves as a foundation for the subsequent generation of synthetic samples. We then collect a dataset of synthetic data, $\mathcal{D}_{\text{syn}}$,

by conditioning the pre-trained diffusion model on text prompts formatted as `class-label` and random images from the corresponding class. We leverage feedback signals from the pre-trained classifier to guide the sampling process of the pre-trained diffusion model, promoting useful samples for $f_\phi$. Finally, we train the classifier from scratch on the union of the original real data and the generated synthetic data, $\mathcal{D}_{\text{real}} \cup \mathcal{D}_{\text{syn}}$.

## 3.1 Increasing the Usefulness of Synthetic Data: Feedback-guided Synthesis

**Feedback-guided synthesis.** We propose feedback-guidance, a modification of the diffusion models' standard sampling process for generating *useful* samples for training a classifier by getting feedback from the classifier itself. While standard classifier-guidance in diffusion models (Dhariwal & Nichol, 2021; Ho et al., 2020) focuses on high-density regions, our method prioritizes the generation of *useful* samples, by leveraging feedback from our pre-trained classifier $f_\phi$. Leveraging classifier feedback allows for a systematic approach for generating useful samples that provide gradient for the classification task at hand.

Our proposed feedback-guidance might be reminiscent of classifier-guidance in diffusion models (Dhariwal & Nichol, 2021; Ho et al., 2020), which drives the sampling process of the generative model to produce images that are close to the distribution modes. The proposed feedback-guidance is also related to the literature aiming to increase sample diversity in diffusion models (Sehwag et al., 2022; Um & Ye, 2023), whose goal is to drive the sampling process of the generative model towards low density regions of the learned distribution. Instead, the goal of our proposed feedback-guidance is to synthesize samples which are *useful* for a classifier to improve its performance. Furthermore, our framework is inspired by the literature in active learning (Wang et al., 2016; Wu et al., 2022), where useful samples are selected from a pool of samples, whereas we propose to *generate* useful samples.

Formally, let $\mathcal{D}_{real}$ be a training dataset of real data, $f_\phi$ a classifier, and $G_\theta$ a state-of-the-art pre-trained diffusion model. We start by training $f_\phi$ on $\mathcal{D}_{real}$, and define $h \in \{0, 1\}$ as a binary variable that describes whether a sample is useful for the classifier $f_\phi$ or not. Our goal is to generate samples from a specific class that are informative, *i.e.* from the distribution of $p(x|h, y)$. To generate samples using the reverse sampling process defined in section 2, we need to compute $\nabla_x \log p(x|h, y)$. Following Eq. 2, we have:

$$\nabla_x \log \hat{p}_{\gamma,\omega}(x|h, y) = \nabla_x \log \hat{p}_\theta(x) + \gamma \nabla_x \log \hat{p}_\theta(y|x) + \omega \nabla_x \mathcal{C}(x, y, f_\phi), \tag{3}$$

where $\mathcal{C}(x, y, f_\phi)$ is a criterion function approximating the sample usefulness ($h$), and $\omega$ is a scaling factor that controls the strength of the signal from our criterion function. Note that by using a pre-trained diffusion model, we have access to the estimated class conditional score function $\nabla_x \log \hat{p}_\theta(x|y)$ as well as the estimated unconditional score function $\nabla_x \log \hat{p}_\theta(x)$. The derivation of Eq. 3 is presented in Appendix A.1.

### 3.1.1 Feedback Criteria $\mathcal{C}(x, y, f_\phi)$

We explore criteria functions that promote generating samples which are informative and challenging for the classifier. We consider three feedback criteria: (1) the classifier's loss on the generated samples; (2) the classifier's prediction entropy on the generated samples; and (3) the hardness score (Sehwag et al., 2022).

**Classifier Loss.** To focus on generating samples that pose a challenge for the classifier $f_\phi$, we use the classifier's loss as the criterion function for the feedback guided sampling. Formally, we define $\mathcal{C}(x, y, f_\phi)$ in terms of the loss function $\mathcal{L}$ as:

$$\mathcal{C}(x, y, f_\phi) = \mathcal{L}(f_\phi(x), y). \tag{4}$$

Since $\mathcal{L}$ is the negative log-likelihood, and following Eq. 3, we have:

$$\nabla_x \log \hat{p}_\omega(x|h, y) = \nabla_x \log \hat{p}_\theta(x) + \gamma \nabla_x \log \hat{p}_\theta(y|x) - \omega \nabla_x \log \hat{p}_\phi(y|x). \tag{5}$$

Note that $\hat{p}_\theta$ is the probability distribution modeled by the generative model and $\hat{p}_\phi$ is the probability distribution modeled by the classifier $f_\phi$. We are effectively moving towards space where under the classifier $f_\phi$ the samples have **lower** probability[2], but simultaneously the term $\nabla_x \log \hat{p}_\theta(y|x)$ which is modeled by the generative model, ensures that the samples belong to class $y$.

---

[2]This is in contrast to classifier guidance, which directs the sampling process towards examples that have a high probability under class $y$.

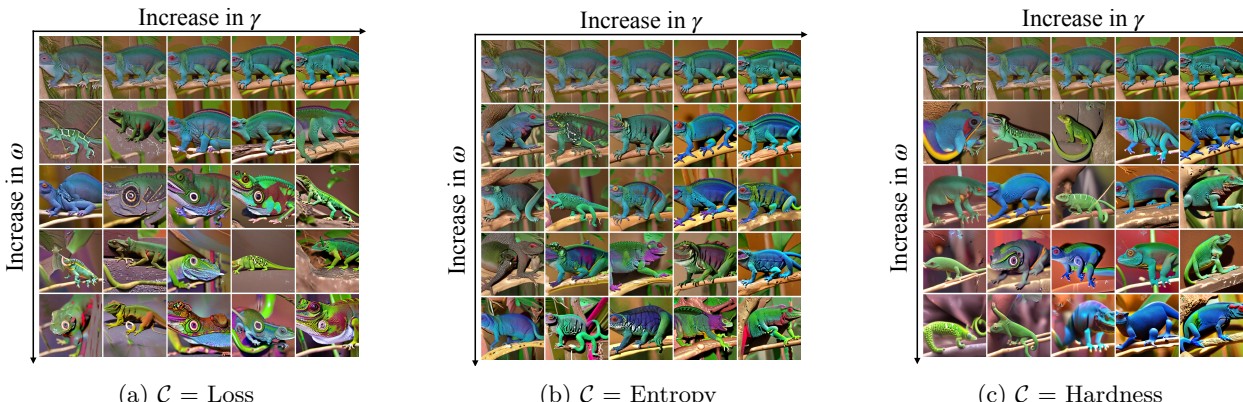

(a) $\mathcal{C} = $ Loss          (b) $\mathcal{C} = $ Entropy          (c) $\mathcal{C} = $ Hardness

Figure 3: A grid of images generated for the class "African Chameleon" using different criterion for feedback guidance. Along the x-axis, from left to right conditional guidance scale or the $\gamma$ in Eq. 3 is increased, while along the y-axis, from top to bottom, the classifier-feedback guidance scale or the $\omega$ in Eq. 3 is increased. The random seed is consistent across all images, ensuring that any observed variations are only due to changes in the guidance scales. Moving from left to right, it is evident that increasing $\gamma$ results in samples that are more *faithful* to the "African Chameleon" class. However, this comes at the cost of generating very typical, easily classifiable images. Conversely, as we move from top to bottom, increasing the classifier-feedback guidance results in the generation of more 'challenging' or atypical images of African chameleon. However, this may come at the cost of moving away from the class "African Chameleon", thus it is important to tune $\gamma$ and $\omega$ to generate useful samples.

**Entropy.** Another measure for the usefulness of the generated samples is the entropy (Shannon, 2001) of the output class distributions for $x$ predicted by $f_\phi(x)$. Entropy is a common measure that quantifies the uncertainty of the classifier on a sample $x$ (Wang et al., 2016; Sorscher et al., 2022; Simsek et al., 2022). We adopt entropy as a criterion, $\mathcal{C} = H(f_\phi(x))$, as higher entropy leads to generating more informative samples. Following Eq. 3, we have,

$$\nabla_x \log \hat{p}_\omega(x|h,y) = \nabla_x \log \hat{p}_\theta(x) + \gamma \nabla_x \log \hat{p}_\theta(y|x) + \omega \nabla_x H(f_\phi(x)). \tag{6}$$

This sampling method encourages the generation of samples for which the classifier $f_\phi$ has low confidence in its predictions.

**Hardness Score.** The Hardness score (Sehwag et al., 2022) quantifies how difficult or informative a sample $(x,y)$ is for a given classifier $f_\phi$. It is defined as:

$$\mathcal{HS}(x,y,f_\phi) = \frac{1}{2}\Big[\big(f_\phi(x) - \mu_y\big)^T \Sigma_y^{-1}\big(f(x) - \mu_y\big) + \ln\big(\det(\Sigma_y)\big) + k\,\ln(2\pi)\Big], \tag{7}$$

where $\mu_y$ and $\Sigma_y$ are the sample mean and sample covariance for embeddings of class $y$ and $k$ is the dimension of embedding space. We directly adopt the *Hardness* score as a criterion;

$$\nabla_x \log \hat{p}_\omega(x|h,y) = \nabla_x \log \hat{p}_\theta(x) + \gamma \nabla_x \log \hat{p}_\theta(y|x) + \omega \nabla_x \mathcal{HS}(x,y,f_\phi). \tag{8}$$

This sampling procedure promotes generating samples that are challenging for the classifier $f_\phi$.

To showcase the interplay between $\gamma$ and $\omega$ and how changing the criterion function $\mathcal{C}$ affects the sampling process, we plot grids of synthetically generated images in Figure 3. All samples are conditioned to generate an "African Chameleon", with variations in feedback-guidance scales based on three different criteria: entropy, loss, and hardness. $\gamma$ serves the role of ensuring that the generated images are faithful to the visual characteristics of an "African Chameleon", such as color patterns, skin details, or posture. On the other hand, $\omega$ relies on the uncertainties in the classifier's predictions to guide the generative model.

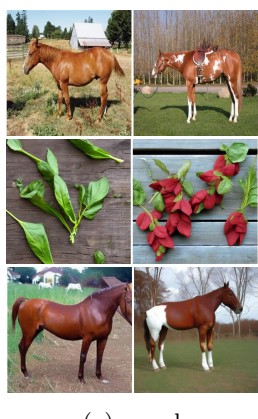
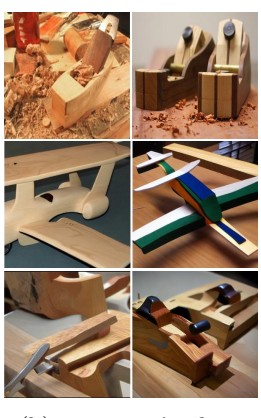
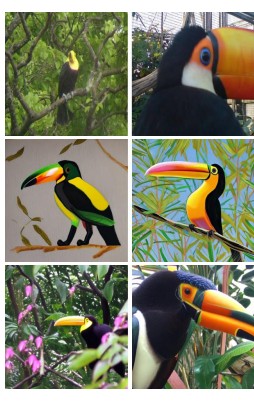

(a) sorrel                   (b) carpenter's plane                   (c) toucan

Figure 4: In each figure: first row depicts samples from Imagenet-LT, second row shows samples from LDM conditioned on text prompt (class label), and third row displays samples generated by conditioning the LDM on both text prompt and the image embedding of a random real sample from the class. **(a) Homonym ambiguity**: the text prompt *sorrel* refers to either the sorrel horse or the sorrel plant. However, in the Imagenet-LT dataset class sorrel refers to the horse while LDM generates the plant. **(b) Text misinterpretation**: LDM may interpret the text prompt incorrectly. For example the class *carpenter's plan* refers to a carpentering tool, while LDM generates wooden planes. **(c) Stylistic Bias**: When prompted with *Toucan*, LDM mostly generates drawings of the bird toucan. However, in the training distribution, we mostly observe real birds. Conditioning on a random training image allows the LDM to generate samples with the correct style. See Section 3.2 and also Figures 7, 8.

### 3.1.2  Feedback-guided Synthesis in Latent Diffusion Models

To apply feedback-guided synthesis in latent diffusion models (LDMs), we need to compute the criteria function $\mathcal{C}(f_\phi(x_t))$ at each step of the reverse sampling process with the minor change that the diffusion is applied on the latent variables $z$. However, the classifier $f_\phi$ operates on the pixel space $x$. Consequently, a naive implementation of feedback-guided sampling would require a full reverse chain to find $z_0$, which would then be decoded to find $x_0$ to finally compute $\mathcal{C}(f(x_0))$. Therefore, to reduce the computational cost, instead of applying the full reverse chain, we use the DDIM *predicted* $z_0$ (or equivalently predicted $x_0$ in Eq. 1) at each step of the reverse process. We find that this approach is computationally much cheaper and is highly effective.

### 3.2  Towards Synthetic Generations Lying within the Distribution of Real Data

We identify three scenarios where using only text prompt results in synthetic samples that are not close to the real data used to train machine learning downstream models:

- **Homonym ambiguity.** A single text prompt can have multiple meanings. For example, consider generating data for the class `sorrel` that could either refer to the sorrel horse or the sorrel plant. See Figure 4 (a) for further illustrations.
- **Text misinterpretation.** The text-to-image generative model can produce images which are semantically inconsistent or partially consistent with the input prompt. An example of that is the class `carpenter's plane` from ImageNet-LT. When prompted with this term, the diffusion model generated images of wooden planes instead of the intended carpentry tools. See Figure 4 (b) for further illustrations.
- **Stylistic Bias.** The generative model can produce images with a particular style for some prompts, which does not match the style of the real data. For instance, the `Toucan` images in the ImageNet-LT dataset are mostly real photographs, but the generative model frequently outputs drawings of this species of bird. See Figure 4 (c) for further illustrations. Also see more samples in Figures 7, 8.

**Conditioning on:**
text prompt: *"Persian cat"* +
Image embedding of a real sample

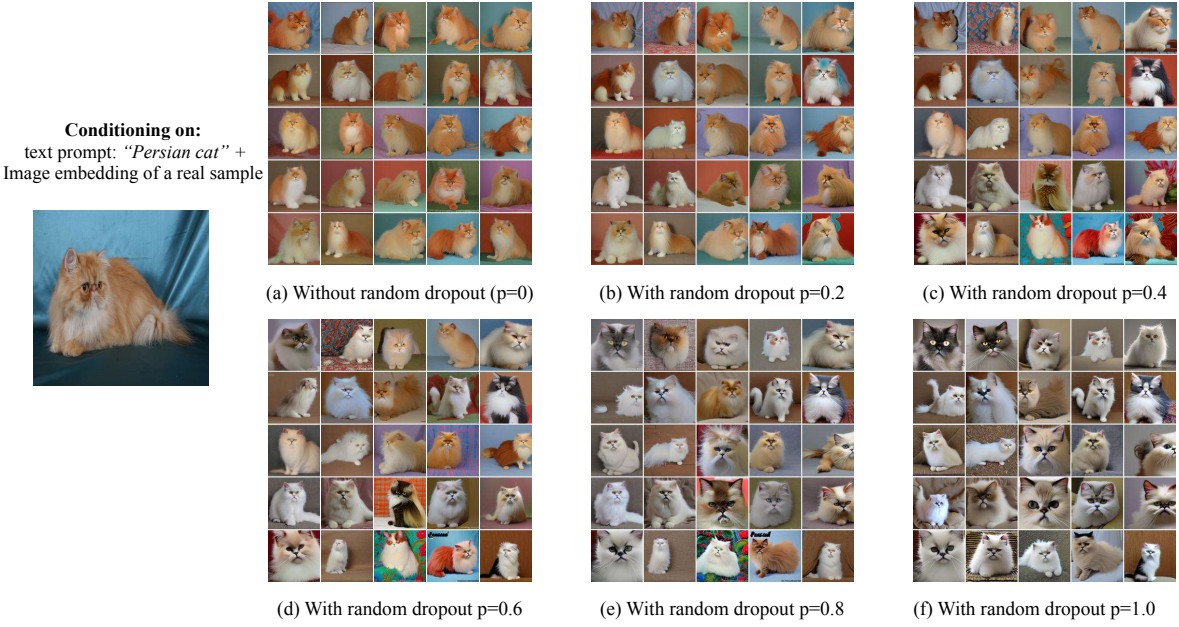

(a) Without random dropout (p=0)  (b) With random dropout p=0.2  (c) With random dropout p=0.4

(d) With random dropout p=0.6  (e) With random dropout p=0.8  (f) With random dropout p=1.0

Figure 5: Synthetic sample generation using text prompt and image embedding. We plot different levels of dropout on the image embedding. We condition on a single random real sample (plotted on the left most). As observed, using only image conditioning and text (a), we observe very low diversity in the generations. As we increase the dropout probability, we observe more diversity. If we only condition on the text prompt (f), we also observe low diversity.

To alleviate the above-described issues, we borrow from the generative models' literature a dual-conditioning technique (Mishra et al., 2023). In this approach, the generator is conditioned on both a text descriptor containing the class label and a randomly selected real image from the same class in the real training dataset. This additional layer of conditioning steers the diffusion model to generate samples which are more similar to those in the real training data; see Figure 4, to contrast samples from text-conditional models with those of text-and-image-conditional models. Using the unCLIP model (Rombach et al., 2022), the noise prediction network $\epsilon_\theta^{(t)}(\mathbf{x}_t)$ in Equation 1 is extended to be a function of the conditioning image's embedding, denoted as $\epsilon_\theta^{(t)}(\mathbf{x}_t, \mathbf{z}_{\text{cond}})$, where $\mathbf{z}_{\text{cond}}$ is the CLIP embedding of the conditioning image.

### 3.3 Increasing the Conditional Diversity of Synthetic Data

As discussed in Section 3.2 leveraging conditioning from real images to synthesize data results in generating samples that are closer to the real data distribution. However, this comes at the cost of limiting the generative model's ability to produce diverse images. Yet, such diversity is essential to train downstream classification models. We propose to apply random dropout on image embedding. Dropout is a technique used for preventing over-fitting by randomly setting a fraction of input units to 0 at each update during training time (Srivastava et al., 2014). In this setup, the application of dropout serves a different yet equally crucial purpose: enhancing the diversity of generated images.

By applying random dropout to the embedding of the conditioning image, we effectively introduce variability into the information that guides the generative model. This stochasticity breaks the deterministic link between the conditioning image and the generated sample, thereby promoting diversity in the generated images. For instance, if the conditioning image contains a `Persian Cat` with a specific set of features (*e.g.,* shape, color, background), dropout might nullify some of these features in the embedding, leading

Table 1: Classification accuracy on ImageNet-LT. All models are trained using ResNext50 backbone, unless marked with **. Models marked with ** use a ViT-B16 architature and are additionally initialized with CLIP (Radford et al., 2021) weights.

| Method | # Syn. data | ImageNet-LT | | | |
| --- | --- | --- | --- | --- | --- |
| | | Overall | Many | Medium | Few |
| ERM (Park et al., 2022) | ✗ | 43.43 | 65.31 | 36.46 | 8.15 |
| Decouple-cRT (Kang et al., 2020) | ✗ | 47.3 | 58.8 | 44.0 | 26.1 |
| Decouple-LWS (Kang et al., 2020) | ✗ | 47.7 | 57.1 | 45.2 | 29.3 |
| Remix (Chou et al., 2020) | ✗ | 48.6 | 60.4 | 46.9 | 30.7 |
| Balanced Softmax (Ren et al., 2020) | ✗ | 51.0 | 60.9 | 48.8 | 32.1 |
| CMO (3 experts) (Park et al., 2022) | ✗ | 56.2 | 66.4 | 53.9 | 35.6 |
| Mix-Up GLMC (Du et al., 2023) | ✗ | 57.21 | 64.76 | 55.67 | 42.19 |
| VL-LTR** (Tian et al., 2022) | ✗ | **77.2** | **84.5** | **74.6** | **59.3** |
| Fill-Up (Shin et al., 2023) | 2.6M | 63.7 | 69.0 | 62.3 | 54.6 |
| LDM (txt) | 1.3M | 57.90 | 64.77 | 54.62 | 50.30 |
| LDM (txt and img) | 1.3M | 58.92 | 56.81 | 64.46 | 51.10 |
| LDM-FG (Loss) | 1.3M | 60.41 | 66.14 | 57.68 | 54.1 |
| LDM-FG (Hardness) | 1.3M | 56.70 | 58.07 | 55.38 | 57.32 |
| LDM-FG (Entropy) | 1.3M | 64.7 | 69.8 | 62.3 | 59.1 |
| LDM-FG (Entropy) + VL-LTR** | 1.3M | **77.27** | **83.59** | **75.57** | **64.31** |

the generative model to explore other plausible variations of `Persian Cat`. Intuitively, this diverse set of generated samples, which now contain both the core characteristics of the class and various incidental features, better prepares the downstream classification models for real-world scenarios where data can be highly heterogeneous. See Figure 5.

## 4 Experiments

We explore two classification scenarios: (1) class-imbalanced classification, where we generate synthetic samples to create a balanced representation across classes; and (2) group-imbalanced classification, where we leverage synthetic data to achieve a balanced distribution across groups.

### 4.1 ImageNet-LT: Class-imbalanced Classification

**Dataset.** The ImageNet-LongTail (ImageNet-LT) dataset (Liu et al., 2019) is a subset of the original ImageNet (Deng et al., 2009) consisting of 115.8K images distributed non-uniformly across 1,000 classes. In this dataset, the number of images per class ranges from a minimum of 5 to a maximum of 1,280. However, the test and validation sets are balanced. In line with related literature (Shin et al., 2023), our goal is to synthesize missing data points in a way that, when combined with the real data, results in a uniform distribution of examples across all classes.

**Experimental setup.** We leverage the pre-trained state-of-the-art image-and-text conditional LDM v2-1-unclip (Rombach et al., 2022) to sample from. We adopt the widely used ResNext50 architecture as well as the ViT-B16 model as the classifiers for our experiments on ImageNet-LT. For ResNext50, our classifier is trained for 150 epochs, for ViT-B models, the classifier is trained with real data for a total of 100 epochs and then fine-tuned using real and synthetic data for another 10 epochs. To improve model scaling with synthetic data, we modify the training process to include 50% real and 50% synthetic samples in each mini-batch. [3] We apply a balanced mini-batch approach when training all LDM methods. We also use the balanced Softmax (Ren et al., 2020) loss when training the classifier. For additional experimental details see Appendix F.

---

[3]This change boosts the performance by nearly 4 points on ResNext50.

**Metrics.** We follow the standard protocol and report the overall average accuracy across classes, and the stratified accuracy across classes *Many* (any class with over 100 samples), *Medium* (any class with 100-20 samples), and *Few* (any class with less than 20 samples).

**Baselines.** We compare the proposed approach with prior art which does not leverage synthetic data from pre-trained generative models and with recent literature which does. Furthermore, we compare against methods that leverage pretrained CLIP weights (Radford et al., 2021). We also compare the proposed approach with a vanilla sampling LDM that uses only the text prompts[4] as conditioning. We report the results of our proposed framework for the three feedback guidance techniques introduced in section 3, namely, Loss, Hardness and Entropy. When leveraging synthetic data, we balance ImageNet-LT by generating as many samples as required to obtain 1,300 examples per class.

**Discussion.** Table 1 presents the results, see further analysis on the performances of the model as the number of synthetic data scales in Appendix D. We observe that approaches which do not leverage generated data exhibit the lowest accuracies across all groups, and overall, with Mix-Up obtaining the best results. It is worth noting that Mix-Up leverages advanced data augmentation strategies on the real data, which result in new samples. When comparing frameworks which use generated data from state of the art generative models, our framework surpasses the LDM baseline. Notably, the LDM with Feedback-Guidance (LDM-FG) based on the entropy criteria increases the LDM baseline performance $\sim 5$ points overall and, perhaps more interestingly, these improvements translate into a $\sim 9$ points boost on classes Few. Our best LDM-FG also surpasses the most recent competitor, Fill-Up (Shin et al., 2023), by 1% accuracy point overall while using a half the amount of synthetic images. This highlights the importance of generating samples that are close to the real data distribution—as Fill-Up does— but also improving their diversity and usefulness. Furthermore, we showcase that since our method is data-driven, it can be combined with any existing algorithmic or architectural approaches. Specifically, we compare against recent benchmarks that leverage visual-linguistic long-tailed recognition along with initializing from CLIP weights (Tian et al., 2022). We observe that additional synthetic data with Feedback Guidance using entropy outperforms the baseline VL-LTR specifically we improve the performance on classes few by 5 points and on classes medium by 1 point.

## 4.2 Places-LT: Class-imbalanced Classification

**Dataset.** To further study the effect of feed-back guidance on different datasets, we study the Places-Long tailed dataset (Liu et al., 2019). This dataset consists of 365 classes where the minimum number of examples in a class is 5 and the maximum is 4980. However, test and validation sets are balanced across classes. Similar to ImageNet-LT, we aim to synthesize data points in a way that, when combined with the real data, results in a uniform distribution across all classes.

**Experimental setup.** We leverage the pre-trained state-of-the-art image-and-text conditional LDM v2-1-unclip (Rombach et al., 2022) to sample from. We follow the common practice in the literature (Kang et al., 2020; Liu et al., 2019) and use a pretrained ResNet-152. We apply two stages of training (Ren et al., 2020). Stage one where the model is trained for 30 epochs. Once the base model is obtained, we further fine-tune the last layer of the model for 10 epochs using Meta Sampler and Balanced Softmax (Ren et al., 2020).

**Metrics.** Similar to ImageNet-LT, we report the overall average accuracy as well as the stratified accuracy across classes *Many* (any class with over 100 samples), *Medium* (any class with 100-20 samples), and *Few* (any class with less than 20 samples).

**Baselines.** We compare against previous methods that do not use synthetic data as well as the recently proposed Fill-Up (Shin et al., 2023) that uses synthetic data. We achieve state of the art results overall as well as on classes medium, and few. When leveraging synthetic data, we balance Places-LT by generating as many samples as required to obtain 4980 examples per class.

**Discussion.** Table 2 summarizes the results. We observe that using synthetic data increases the overall performance. Comparing Fill-up and Feedback Guidance, we see that using any type of feedback guidance greatly boosts the performance on classes Few with Entropy out-performing our feedback criteria.

---

[4]Text prompts are in the format of `class-label`.

Table 2: Classification accuracy on Places-LT using ResNet-152.

| Method | # Syn. data | Places-LT | | | |
|---|---|---|---|---|---|
| | | Overall | Many | Medium | Few |
| ERM (Cui et al., 2021) | ✗ | 30.2 | 45.7 | 27.3 | 8.2 |
| Decouple-LWS (Kang et al., 2020) | ✗ | 37.6 | 40.6 | 39.1 | 28.6 |
| Balanced Softmax (Ren et al., 2020) | ✗ | 38.6 | 42.0 | 39.3 | 30.5 |
| ResLT (Cui et al., 2022) | ✗ | 39.8 | 39.8 | 43.6 | 31.4 |
| MiSLAS (Zhong et al., 2021) | ✗ | 40.4 | 39.6 | 43.3 | 36.1 |
| Fill-Up (Shin et al., 2023) | 1.8M | 42.6 | **45.7** | 43.7 | 35.1 |
| LDM-FG (Loss) | 1.8M | 42.3 | 41.2 | 44.4 | 39.8 |
| LDM-FG (Hardness) | 1.8M | 41.8 | 40.7 | 44.2 | 38.5 |
| LDM-FG (Entropy) | 1.8M | **42.8** | 41.7 | **44.9** | **40.0** |

## 4.3 NICO++: Group-imbalanced Classification

**Dataset.** We follow the sub-population shift setup of NICO++(Zhang et al., 2023; Yang et al., 2023) which contains 62,657 training examples, 8,726 validation and 17,483 test examples. This dataset contains 60 classes of animals and objects within 6 different contexts (autumn, dim, grass, outdoor, rock, water). The pair of class-context is called a *group*, and the dataset is imbalanced accross groups. In the training set, the maximum number of examples in a group is 811 and the minimum is 0. For synthetic samples generated using our framework see Figure 11.

**Experimental setup.** We again leverage the pre-trained state-of-the-art image-and-text conditional LDM v2-1-unclip (Rombach et al., 2022) as high performant generative model to sample from. Since some groups in the dataset do not contain any real examples, we cannot condition the LDM model on random images from group, and so instead, we condition the LDM on random in-class examples. We adopt the ResNet50 (He et al., 2016) architecture as the classifier, given its ubiquitous use in prior literature. For each baseline, we train the classifier with five different random seeds.

**Metrics.** Following prior work on sub-population shift (Yang et al., 2023; Sagawa et al., 2019), we report worst-group accuracy (WGA) as the benchmark metric. We also report overall accuracy.

**Baselines.** We compare our framework against the NICO++ benchmarks (Zhang et al., 2023; Yang et al., 2023). These state-of-the-art methods may leverage data augmentation but do not rely on synthetic data from generative models. We also consider a vanilla LDM baselines conditioned on text prompt, and report results for all three criteria. We generate the text prompts as `class-label in context`. We balance the NICO++ dataset such that each group has 811 samples, overall we generate 229k samples.

**Discussion.** Table 3 presents the average performance across five random seeds of our method in contrast with previous works. As shown in the table, our method achieves remarkable improvements over prior art which does not leverage synthetic data from generative models. More precisely, we observe notable WGA improvements of $\sim 6\%$ over the best previously reported results on the ResNet architecture. When comparing against baselines which do leverage synthetic data from state-of-the-art generative models, our framework remains competitive in terms of average accuracy, and importantly, surpasses the ERM baseline in terms of WGA by over 14%. The improvements over LDM further emphasize the benefit of generating samples that are close to the real data distribution, are diverse and useful when leveraging generated data for representation learning. Notably, the improvements achieved by our framework are observed for all the explored criteria, although entropy consistently yields the best results.

Also note that our framework tackles the group-imbalanced problem from the data perspective. Any algorithmic approach such as Bsoftmax (Ren et al., 2020), IRM (Arjovsky et al., 2019), or GroupDRO (Sagawa et al., 2019) can be applied on top of our approach.

Table 3: Classification average and worst group accuracy on NICO++ dataset using ResNet50 pretrained on Imagenet.

| Algorithm | # Syn. data | Avg. Accuracy | Worst Group Accuracy |
|---|---|---|---|
| ERM (Yang et al., 2023) | ✗ | $85.3 \pm 0.3$ | $35.0 \pm 4.1$ |
| Mixup (Zhang et al., 2017) | ✗ | $84.0 \pm 0.6$ | $42.7 \pm 1.4$ |
| GroupDRO (Sagawa et al., 2019) | ✗ | $82.2 \pm 0.4$ | $37.8 \pm 1.8$ |
| IRM (Arjovsky et al., 2019) | ✗ | $84.4 \pm 0.7$ | $40.0 \pm 0.0$ |
| LISA (Yao et al., 2022) | ✗ | $84.7 \pm 0.3$ | $42.7 \pm 2.2$ |
| BSoftmax (Ren et al., 2020) | ✗ | $84.0 \pm 0.5$ | $40.4 \pm 0.3$ |
| CRT (Kang et al., 2019) | ✗ | $85.2 \pm 0.3$ | $\mathbf{43.3} \pm 2.7$ |
| LDM | 229k | $86.02 \pm 1.14$ | $32.66 \pm 1.33$ |
| LDM FG (Loss) | 229k | $84.55 \pm 0.20$ | $45.60 \pm 0.54$ |
| LDM FG (Hardness) | 229k | $84.66 \pm 0.34$ | $40.80 \pm 0.97$ |
| LDM FG (Entropy) | 229k | $85.31 \pm 0.30$ | $\mathbf{49.20} \pm 0.97$ |

Table 4: Ablation of our framework based on LDM. Results are computed *w.r.t.* the real balanced validation set of the ImageNetLT. All hyper-parameters for each setup are tuned.

| Text | Img | dropout | FG | FID↓ | Density ↑ | Coverage↑ | Avg. / Few validation acc. ↑ |
|---|---|---|---|---|---|---|---|
| ✓ | ✗ | ✗ | ✗ | 18.46 | 0.962 | 0.690 | 59.52 / 50.74 |
| ✓ | ✓ | ✗ | ✗ | 14.24 | 1.019 | 0.676 | 59.95 / 49.5 |
| ✓ | ✓ | ✓ | ✗ | 13.63 | 1.063 | 0.722 | 60.16 / 54.79 |
| ✓ | ✓ | ✓ | Loss | 18.48 | 0.867 | 0.639 | 62.19 / 54.78 |
| ✓ | ✓ | ✓ | Hardness | **10.84** | **1.070** | **0.820** | 57.7 / 56.57 |
| ✓ | ✓ | ✓ | Entopy | 21.36 | 0.821 | 0.614 | **65.7 / 57.7** |

## 4.4 Ablations

To validate the effect of dual image-text conditioning, dropout on the image conditioning embedding, and feedback-guidance, we perform an ablation study and report Fréchet Inception Distance (FID) (Heusel et al., 2017), density and coverage (Naeem et al., 2020), and average accuracy overall and on the classes Few. FID and density serve as a proxy to measure how close the generated samples are to the real data distribution. Coverage serves as proxy for diversity, and accuracy improvement for usefulness. FID, density and coverage are computed by generating 20 samples per class and using the ImageNet-LT validation set (20,000 samples) as reference. The accuracies are computed on the ImageNet-LT validation set. As shown in the Table 4, leveraging the vanilla sampling strategy of an LDM conditioned on text only (row 1) results in the worse performance across metrics. By leveraging image and text conditioning simultaneously (row 2), we improve both FID and density, suggesting that generated samples are closer to the ImageNet-LT validation set. When applying dropout to the image embedding (row 3), we observe a positive effect on both FID and coverage, indicating a higher diversity of the generated samples. Finally, when adding feedback signals (rows 4–6), we notice the highest accuracy improvements (comparing to the model trained only on real data) both on average (except for hardness) and on the classes Few, highlighting the importance of leveraging feedback-guidance to improve the usefulness of the samples for representation learning downstream tasks. It is important to note that quality and diversity metrics such as FID, density and coverage may not be reflective of the usefulness of the generated synthetic samples (compare Hardness row with the Entropy row in Table 4).

## 5 Related work

**Synthetic Data for Training.** Synthetic data serves as a readily accessible source of data in the absence of real data. One line of previous works have used graphic-based synthetic datasets that are usually crafted through pre-defined pipelines using specific data sources such as 3D models or game engines (Peng et al.,

2017; Richter et al., 2016; Bordes et al., 2023; Richter et al., 2016; Peng et al., 2017) as synthetic data to train downstream models. Despite their utility, these methods suffer from multiple limitations, such as a quality gap with real-world data. Another set of prior work has considered using pre-trained GANs as data generators to yield augmentations to improve representation learning (Astolfi et al., 2023; Chai et al., 2021); with only moderate success as the generated samples still exhibited limited diversity (Jahanian et al., 2021). More recent studies have begun to leverage pre-trained text-to-image diffusion models for improved classification (He et al., 2023; Sariyildiz et al., 2023; Shipard et al., 2023; Bansal & Grover, 2023; Dunlap et al., 2023) or other downstream tasks (Wu et al., 2023; You et al., 2023; Lin et al., 2023; Tian et al., 2023). For example, Imagen model (Saharia et al., 2022) was fine-tuned by (Azizi et al., 2023) to improve Imagenet (Deng et al., 2009) per-class accuracy. Diffusion classifier (Li et al., 2023), a pre-trained diffusion model adapted for the task of classification. Other work (He et al., 2023) adapted a diffusion model to generate images for fine-tuning CLIP (Radford et al., 2021). Furthermore, it is shown that prompt-engineering can lead to a training data from which transferable representations can be learned(Sariyildiz et al., 2023). These studies suggest that it is possible to use features extracted from synthetic data for several downstream tasks. However, consistent with our findings, other work (He et al., 2023; Shin et al., 2023) argue that these methods may produce noisy, non-representative samples when applied to data-limited scenarios, such as those with long-tailed and open-ended distributions. Use of textual inversion (Dhariwal & Nichol, 2021) in an image synthesis pipeline for long-tailed scenarios for improved domain alignment and classification accuracy was also proposed (Shin et al., 2023).

**Balancing Methods for Imbalanced Datasets.** An effective strategy for mitigating class imbalance include balancing the dataset (Shelke et al., 2017; Idrissi et al., 2022; Shen et al., 2016; Park et al., 2022; Liu et al., 2019). Dataset balancing can either involve up-sampling the minority classes to bring about a uniform class/group distribution or sub-sampling the majority classes to match the size of the smallest class/group. Traditional up-sampling methods usually involve either replicating minority samples or through simple methods such as linearly interpolating between them. However, such simple up-sampling techniques have been found to be less effective in scenarios with limited data (Hu et al., 2015). Sub-sampling is generally more effective, but it carries the risk of overfitting due to reduced dataset size. Another line of research focuses on re-weighting techniques (Sagawa et al., 2019; Samuel & Chechik, 2021; Cao et al., 2019; Ren et al., 2020; Cui et al., 2019). These methods scale the importance of underrepresented classes or groups according to specific criterion, such as their count in the dataset or the loss incurred during training (Liu et al., 2021). Some approaches adapt the loss function itself or introduce a regularization technique to achieve a more balanced classification performance (Ryou et al., 2019; Lin et al., 2017; Pezeshki et al., 2021). Another effective method is the Balanced Softmax (Ren et al., 2020) that adjusts the biases in the softmax layer of the classifier to counteract imbalances in class distribution. Furthermore, to enhance the minority over-sampling, one can augment a variety of minority samples using the extensive context provided by majority class images as a backdrop (Park et al., 2022).

## 6 Conclusion and Discussion

We introduced a framework that leverages a pre-trained classifier together with a state-of-the-art text-and-image generative model to extend challenging long-tailed datasets with *useful, diverse* synthetic samples that are close to the real data distribution, with the goal of improving on downstream classification tasks. We achieved *usefulness* by incorporating feedback signals from the downstream classifier into the generative model; we employed dual image-text conditioning to generate samples that are close to the real data manifold and we improved the *diversity* of the generated samples by applying dropout to the image conditioning embedding. We substantiated the effect of each of the components in our framework through ablation studies. We validated the proposed framework on ImageNet-LT, Places-LT and NICO++, consistently surpassing prior art with notable improvements.

Overall, our framework provides a data approach for imbalanced classification tasks, where accessing real data is expensive or infeasible. Using an off-the-shelf pre-trained diffusion model can be viewed as accessing a compressed form of large-scale data. Additionally, our feedback-guided sampling technique enables the extraction of **useful** information for the task at hand. It is important to note, that our approach is com-

plementary to any existing algorithmic approaches that target imbalanced classification. One can combine both data and algorithmic solutions to a given problem to further improve the results.

**Limitations.** Our approach inherits the limitations of the state-of-the-art generative models, and the image realism and representation diversity are determined by the abilities of the generative model – our guidance mechanism can only explore the data manifold already captured by the model.

**Future work.** In our work, we only consider *one* feedback cycle from a *fully trained* classifier. We leave for future exploration a setup where the feedback from the classifier is being sent to the generator online while the representation learning model is being trained. Furthermore, any differentiable criterion function can be applied in the sampling. For example one can consider an active learning acquisition function as a feedback criterion function (Gal et al., 2017; Cho et al., 2022a).

## 7 Broader Impact Statement

Generative models are known to come with various risks and biases. We highlight that the negative impacts associated with these large scale diffusion models underscore the need for robust ethical guidelines, regulatory oversight, and technological safeguards. In this work, we do not directly address or focus on the biases that may exist in the generations of the diffusion model such as ethical concerns, or societal biases. However, we leverage large-scale pretrained generative models and propose an inference-time intervention to generate useful synthetic examples for imbalanced classification problems. From the classification perspective, we proposed a method aimed at decreasing biases due to long-tail data distribution with a positive impact on model fairness. It is important to note that while our approach contributes to mitigating certain biases, it does not eliminate all potential biases of generative process. We advocate for the continued development of comprehensive ethical frameworks and regulatory measures to address the broader impact of using generative models in different applications. Our work is a step towards improving fairness in machine learning, understanding that it is part of an ongoing effort to build more responsible AI.

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

# A  Appendix

## A.1  Derivation of equation 3

Simply writing the Bayes rule, we have:

$$\nabla_x \log p(x|h, y) = \nabla_x \log \left[ \frac{p(x, h, y)}{p(h, y)} \right] = \nabla_x \log \left[ \frac{p(h|x, y)p(x|y)p(y)}{p(h|y)p(y)} \right]$$
$$= \nabla_x \log \left[ p(h|x, y)p(x|y) \right] = \nabla_x \log p(x|y) + \nabla_x \log p(h|x, y). \tag{9}$$

Note that by using a pre-trained diffusion model, we have access to the estimated class conditional score function $\nabla_x \log \hat{p}_\theta(x|y)$. We then assume there exists a criterion function $\mathcal{C}(x, y, f_\phi)$ that evaluates the usefulness of a sample $x$ based on the classifier $f_\phi$. Consequently, we model $p(h|x, y)$ as:

$$p(h|x, y) = \frac{\exp\left(\mathcal{C}(x, y, f_\phi)\right)}{\mathcal{Z}}, \tag{10}$$

where $\mathcal{Z}$ is a normalizing constant. As a result, we can generate useful samples based on the criteria function $\mathcal{C}(x, y, f_\phi)$, and following Eq. 9, we have:

$$\nabla_x \log \hat{p}_\omega(x|h, y) = \nabla_x \log \hat{p}_\theta(x|y) + \omega \nabla_x \mathcal{C}(x, y, f_\phi), \tag{11}$$

where $\omega$ is a scaling factor that controls the strength of the signal from our criterion function $\mathcal{C}$. Following Eq. 2, we have,

$$\nabla_x \log \hat{p}_\omega(x|h, y) = \nabla_x \log \hat{p}_\theta(x) + \gamma \nabla_x \log \hat{p}_\theta(y|x) + \omega \nabla_x \mathcal{C}(x, y, f_\phi). \tag{12}$$

## A.2  A Toy 2-dimensional Example of Criteria Guidance

Figure 6 illustrates the experimental results of using a simple 2-dimensional dataset for a classification task. The dataset contains two classes represented by blue and red data points. Within each class, the data consists of two modes: the majority mode, containing 90% of the data points, and the minority mode, which holds the remaining 10%. We initially train a diffusion model on this dataset. Sampling from the trained diffusion model generates synthetic data that closely follows the distribution of the original training data, showing an imbalance between the modes of each class (see Figure 6 (b)).

To encourage generation of data from the mode with lower density, we introduce a binary variable $h$ into the model. In this context, $h = 0$ indicates the minority mode, while $h = 1$ signifies the majority mode. Following the criteria guidance discussed in Section 3.1.1, without retraining the model, we modify the sampling process so that the generator is guided towards generating samples of higher entropy. To that end, we train a linear classifier for several epochs until it effectively classifies the majority mode, but the decision boundary intersects the minority mode, resulting in misclassification of those points. Leveraging the classifier's uncertainty around this decision boundary, we guide the generative model to produce higher entropy samples. This results in more synthetic samples being generated from the minority modes of each class (see Figure 6 (c)).

Integrating this synthetic data with the original data produces a more uniform distribution across modes, leading to a more balanced classifier. This is desirable in our context as it mitigates the biases inherent in the original dataset and improves the model's generalization.

# B  Samples Stylistic Bias

Stylistic Bias is one of the challenges in synthetic data generation where the generative model consistently produces images with a particular style for some prompts, which does not correspond to the style of the real data. This results in a mismatch between synthetic and real data, potentially impacting the performance of machine learning models trained on such data.

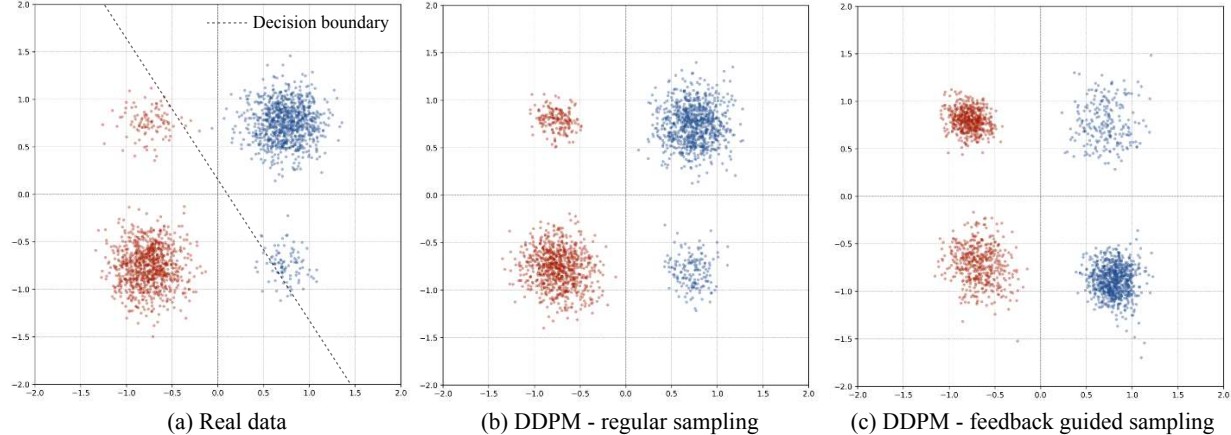

(a) Real data      (b) DDPM - regular sampling      (c) DDPM - feedback guided sampling

Figure 6: Experimental results on a 2-dimensional classification dataset showcasing the effect of feedback-guided sampling. Panel (a): Real data consists of two classes represented by blue and red data points. Within each class, two modes are identified: a majority mode comprising 90% of the data and a minority mode containing the remaining 10%. Panel (b): The synthetic data generated by regular sampling of a DDPM replicates the imbalances of the original dataset. Panel (c): Synthetic data generated after modifying the diffusion model guided by feedback from a linear classifier. Feedback-guided sampling leads to more samples being generated from the minority modes. Combining with real data, it results in more balanced data with increased representation from the minority mode, ultimately improving classifier performance.

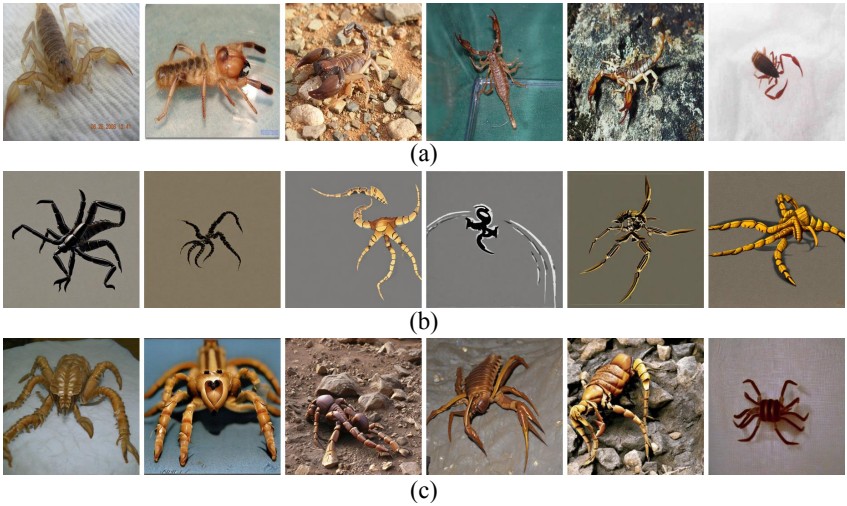

Figure 7: Here we plot more samples from Imagenet-LT where stylistic bias appears in synthetic generations. This scenario arises when the generative model produces images with a particular style for some prompts, which does not match the style of the real data. See Section 3.2 for more details. (a) real samples from class scorpian, (b) synthtic samples using LDM, (c) synthetic samples with image and text conditioning.

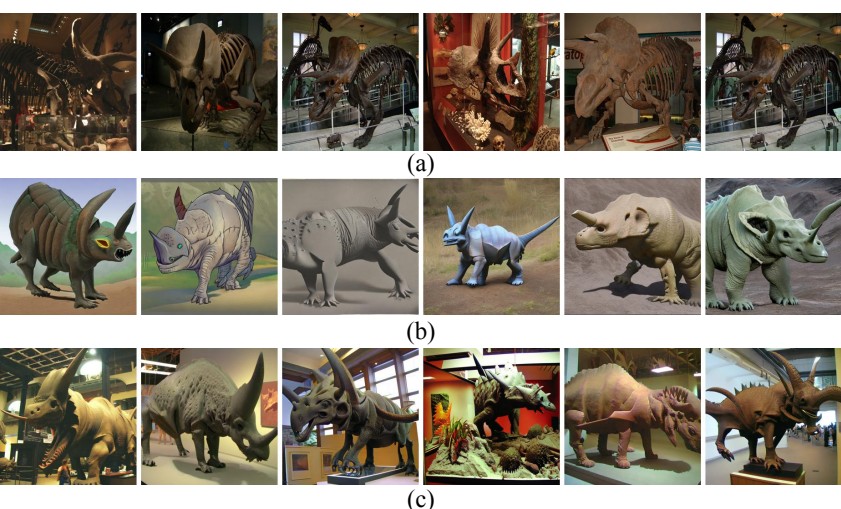

Figure 8: Here we plot more samples from Imagenet-LT where stylistic bias appears in synthetic generations. This scenario arises when the generative model produces images with a particular style for some prompts, which does not match the style of the real data. See Section 3.2 for more details. (a) real samples from class triceratops, (b) synthetic samples using LDM, (c) synthetic samples with image and text conditioning.

## C  Samples of Feedback Guided Sampling

In this section we provide more synthetic samples using Feedback guided sampling. See Figure 9 and 11 for more details.

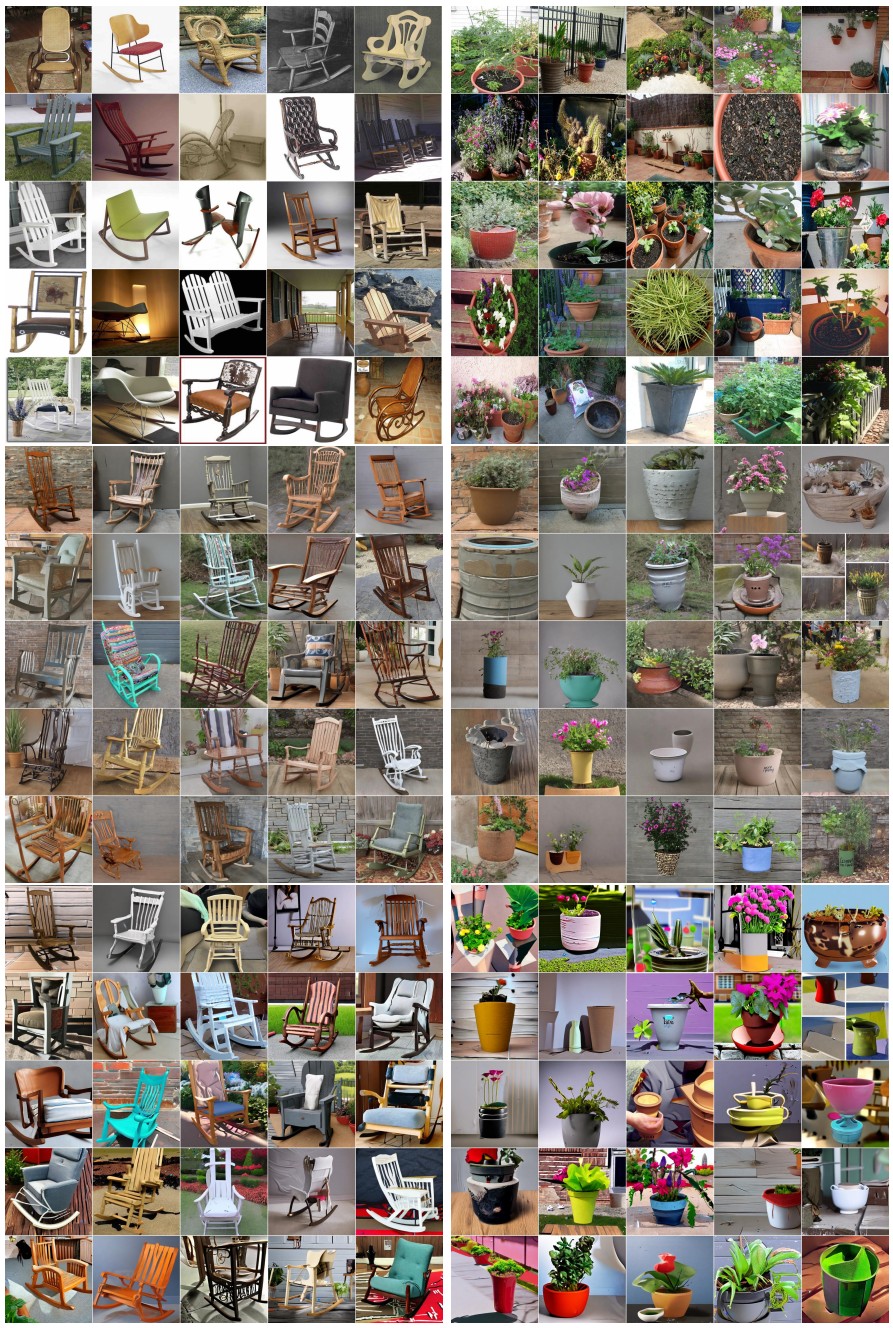

Figure 9: Synthetic samples of two different classes of ImageNet-LT. Column 1: *rocking chair*, Column 2: *flower pot*. First row: Real samples from Imagenet-LT, Second row: synthetic samples vanilla LDM. Third row: synthetic samples using Entropy as the guidance.

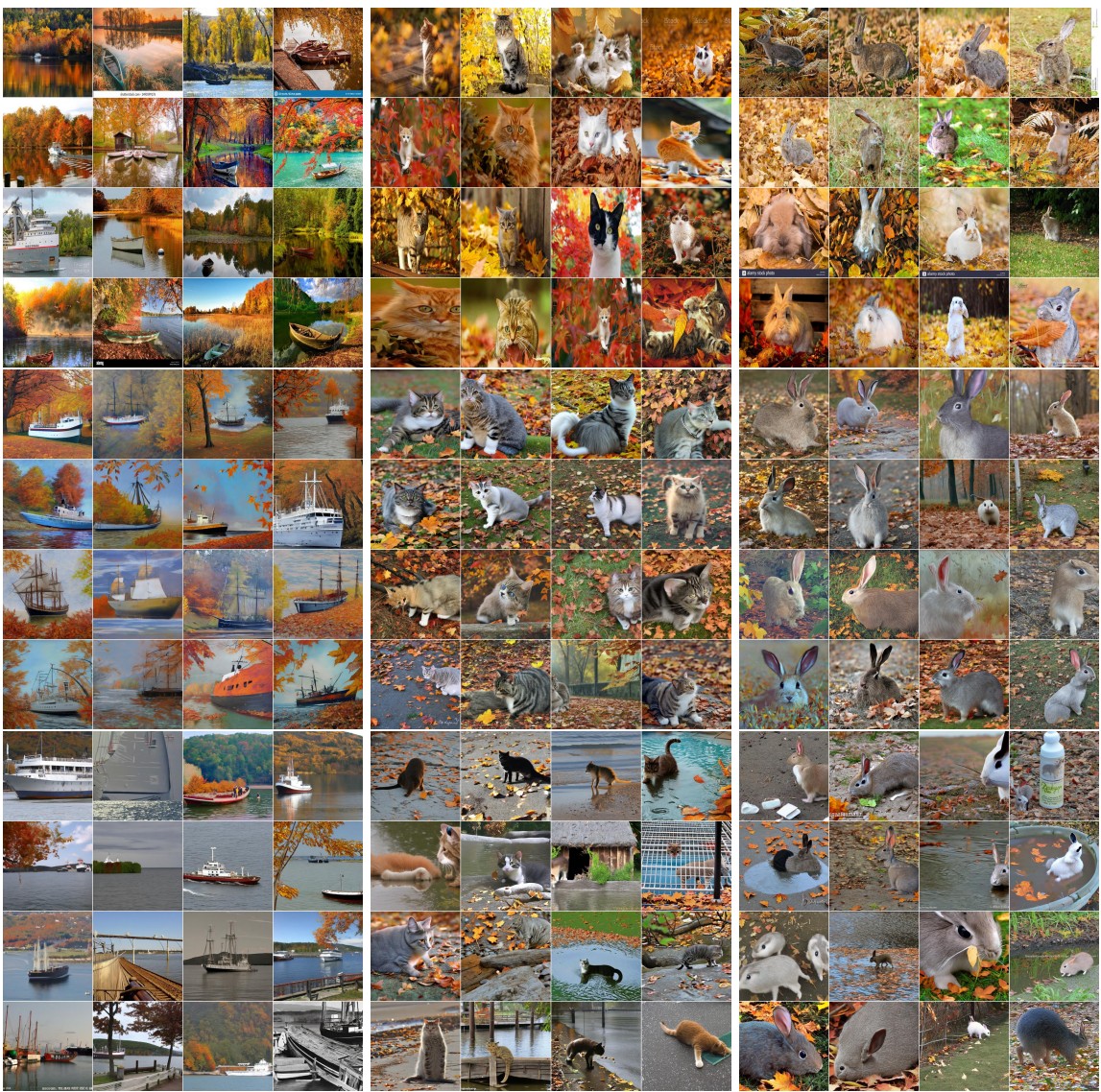

Figure 10: Synthetic samples of three different classes of NICO++. Column 1: class *ship* in context autumn, Column 2: class *cat* in context autumn , Column 3: *class rabbit* in autumn. First row: Real samples from NICO++, Second row: synthetic samples LDM. Third row: synthetic samples using Entropy as the guidance and dropout.

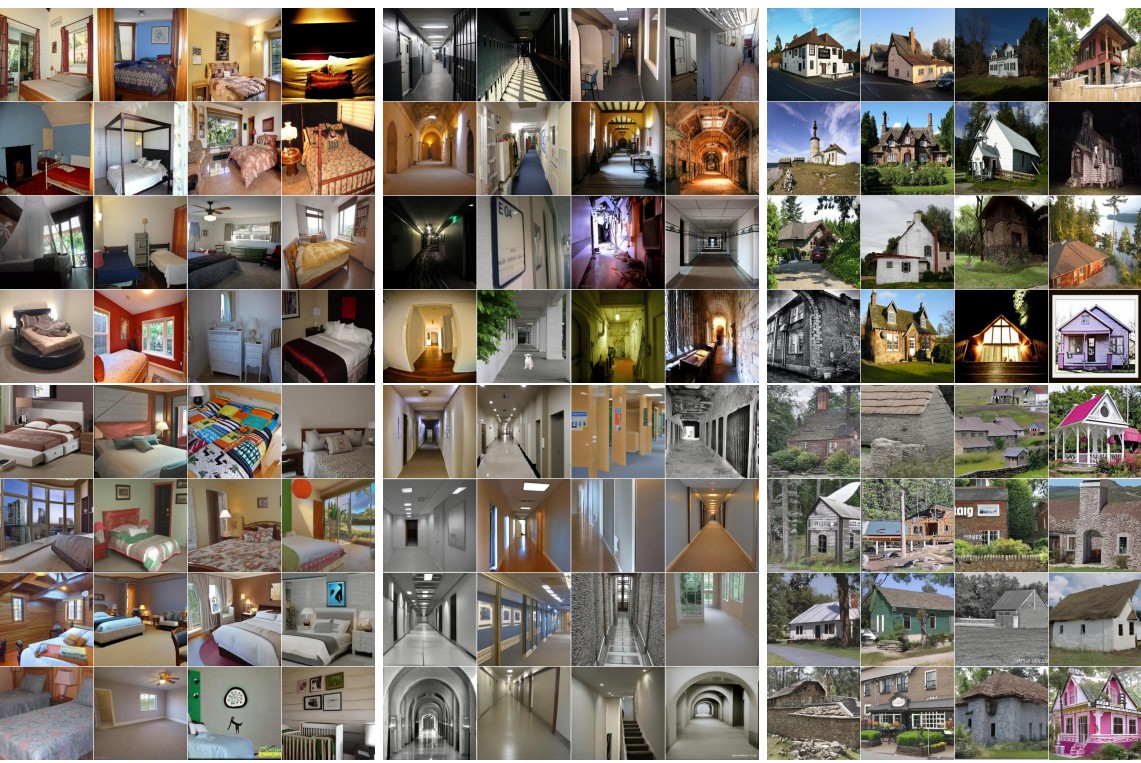

Figure 11: Synthetic samples of three different classes of Places-LT dataset. Column 1: class *bedroom*, Column 2: class *corridor*, Column 3: *class cottage*. First row: Real samples from Places-LT, Second row: synthetic samples using Entropy as the guidance and dropout on image embeddings.

# D    Impact of the number of generated images

Figure 12 shows the performances on a classifier trained on ImageNet-LT when using a different amount of generated images. We show that adding generated synthetic data significantly help to increase the overall performance of the model. In addition, we observe significant gain on the *few* classes which highlight that generated images are well-suited for imbalanced real data scenarios.

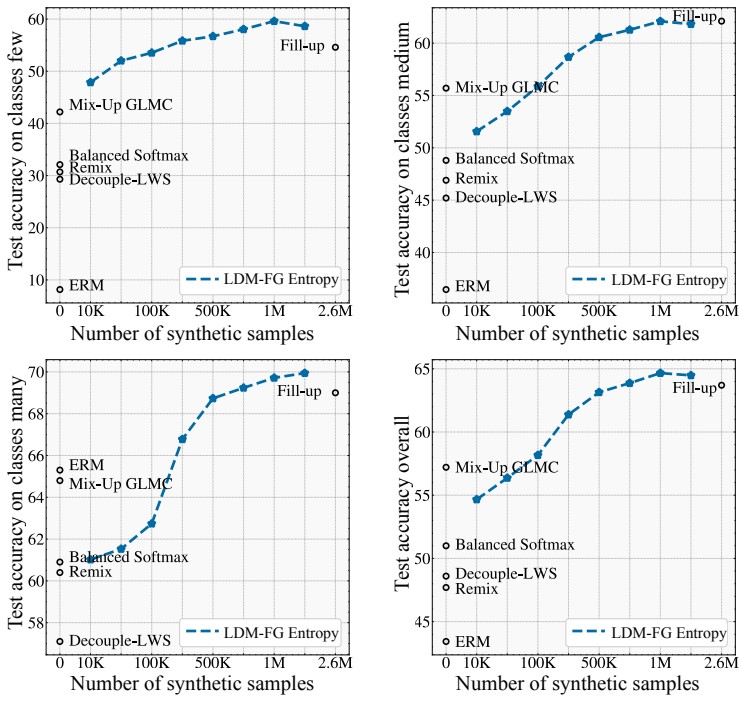

Figure 12: Test accuracy depending on the number of generated synthetic data used to train the classifier. The first curve shows the accuracy on the class Few, while the second one shows the accuracy on the class Medium and the last one shows the accuracy on class Many. Our method significantly outperforms Fill-Up(Shin et al., 2023) while using less synthetic data.

# E    Impact of using balanced softmax with synthetic data

In our experiments, we have used balanced softmax to train our classifier to increase the performances on class Few. We also ran experiments using a weighted average of a traditional cross-entropy loss using balanced softmax with the same loss without balanced softmax. In this experiment, we added a scalar coefficient $\alpha$ which controls the weight of the balanced softmax loss in contrast to the standard loss. In Figure 13, we plot the test accuracy with respect to this balanced softmax weight. Without balanced softmax, the accuracy on the class many is extremely high while the accuracy on class few is much lower. However by increasing the balanced softmax coefficient, we significantly increase the performances on class few and medium as well as the overall accuracy. However, this comes at the price of lower performance for classes Many.

# F    Additional Experiments and Details

**General setup in sample generation**    We use the LDM v2-1-unclip (Rombach et al., 2022) as the state-of-the-art latent diffusion model that supports dual image-text conditioning. We use a pretrained classifier on the real data to guide the sampling process of the LDM. For Imagenet-LT, the classifier is trained using ERM with learning rate of 0.1 (decaying) and weight-decay of 0.0005 and batch-size of 32. For NICO++ we use a pre-trained classifier on Imagnet and then fine-tune it on NICO++.

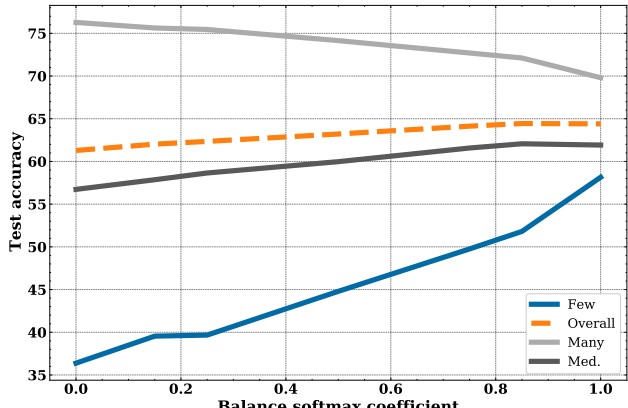

Figure 13: Test accuracy on Few, Medium, Many and Overall with respect to the balanced softmax coefficient. All the models use 1.3M synthetic samples.

We apply 30 steps of reverse diffusion during the sampling. To apply different criteria in the sampling process, we use the pretrained classifier on the real data. For lower computational complexity, we use *float16* datatype in PyTorch. Furthermore, we apply the gradient of the criterion function every 5 steps. So for 30 reverse steps, we only compute and apply the criterion 6 times. Through experiments we find that 5 is optimal as the generated samples look very similar to applying the criterion in every step. For the hardness criterion in Eq. 3.1.1 where we need the $\mu_y$ and $\Sigma_y^{-1}$ for each class, we pre-compute these values. We compute the mean and covariance inverse of the feature representation of the classifier over all real samples. These values are then loaded and used during the sampling process.

### F.1 Computational Cost

In this work, we use state of the art generative models to synthesize samples for an imbalanced classification problem. Since we leverage feedback from the classifier to the generative model, our framework requires a pretrained classifier, once the samples are generated, the classifier is retrained from scratch. Depending on the size of the dataset and the architecture used for the classifier, and the number of hyper-parameters to tune the classifier, the computational cost varies. The generation time for synthetics samples are studied more carefully below. However, the dataset is generated only once and it can be reused for tuning the classifier in the second stage.

For generations, we leverage DDIM sampling and we apply the feedback guidance every 5 steps during the sampling process. We calculate the average wall-clock time (in seconds) per sample generation. Results are reported in Table 5, on an average of 1000 samples, computed on the same model and the same GPU machine (V100) without batch-generation. Faster GPU machines with larger RAM result in faster sample generation.

Note that results are reported per sample generation, one can potentially run the sampling process in parallel. We consider the improvement in the sampling complexity as a future direction of research. Several approaches can be taken to improve the sampling time complexity:

- Using model distillation for diffusion models such as consistency models (Song et al., 2023) or progressive distillation (Salimans & Ho, 2022) that significantly reduce the sampling time.
- Reducing the frequency of guidance to minimize the number of queries to the classifier.
- Reusing the gradients of the classifier guidance criterion in the sampling process during steps that we skip the computation. This can help with reducing the frequency of applying the guidance.

Table 5: Average generation time per sample in seconds. Results are reported on an average of 1000 samples with float16 precision. Values are subject to change on different GPU machines and different computation precision.

| Method | Time (seconds) |
|---|---|
| LDM | $3.819 \pm 0.011$ |
| LDM-FG (Loss) | $21.817 \pm 0.009$ |
| LDM-FG (Hardness) | $21.937 \pm 0.014$ |
| LDM-FG (Entropy) | $21.906 \pm 0.012$ |

### F.2  ImageNet-LT

We follow the setup in previous work (Kang et al., 2019) and use a ResNext50 architecture. We apply the balanced softmax for all the LDM models reported for Imagenet-LT. We train the classifier for 150 epochs with a batch size of 512. We also use standard data augmentations such a random cropping, color-jittering, blur and grayscale during training.

Furthermore, below we provide an ablation study on combining different criteria. Specifically, we study the scenario where each sample is generated with 0.33 probability from each of the three given criteria (Entropy, Loss and Hardness). We upsample the Imagenet-LT dataset to 1.3M samples. As seen in Table 6, entropy outperforms other criteria and the combination of criteria.

Table 6: Classification accuracy on ImageNet-LT using ResNext50 backbone. Ablation on combining different criteria..

| Method | # Syn. data | ImageNet-LT | | | |
|---|---|---|---|---|---|
| | | Overall | Many | Medium | Few |
| LDM-FG (Loss) | 1.3M | 60.41 | 66.14 | 57.68 | 54.1 |
| LDM-FG (Hardness) | 1.3M | 56.70 | 58.07 | 55.38 | 57.32 |
| LDM-FG (Entropy) | 1.3M | **64.7** | **69.8** | **62.3** | **59.1** |
| LDM-FG (Combined) | 1.3M | 62.38 | 67.66 | 59.96 | 56.23 |

### F.3  NICO++

We follow the setup in previous work (Yang et al., 2023), where a pre-trained ResNet50 model on ImageNet is used for all the methods. We assume access to the attributes labels (contexts). For training our LDM model we only apply ERM without any extra algorithmic changes. We use the SGD with momentum of 0.9 and train for 50 epochs. We apply standard data augmentation such as resize and center crop and apply ImageNet statistics normalization. For every method, we try 10 sets of hyper-parameters (learning rate, batch-size[5]) (Yang et al., 2023). We perform model selection and early stopping based on average validation accuracy. We then train the selected model on 5 random seeds and report the test performance.

In Table 7 we provide an ablation analysis on the hyper-parameter $\omega$ that controls the strength of the feedback guidance. The rest of hyper-parameters in the generative model or the classifier are tuned for every set-up. As it can be seen, among the $\omega$ values that we tried, all improved the worst-group-accuracy compared to the case where $\omega = 0$, however the best results are achieved with $\omega=0.03$.

### F.4  Places-LT

We follow the setup in the literature (Kang et al., 2020; Liu et al., 2019) and use a ResNet-152 pretrained on ImageNet. We apply two stages of training (Ren et al., 2020). Stage one where the model is trained for

---

[5]Learning rate is randomly selected from $10^{\text{Uniform}(-4,-2)}$ and batch-size is randomly selected from $2^{\text{Uniform}(6,7)}$.

Table 7: Ablation study on the NICO++ dataset with the entropy as feedback guidance. In each setup, we select the hyper-parameter based on average accuracy on the validation set and report the worst-group accuracy on the test set.

| Feedback guidance coefficient | Worst-group accuracy |
|---|---|
| $\omega = 0$ | $32.66 \pm 1.33$ |
| $\omega = 0.01$ | $45.60 \pm 2.63$ |
| $\omega = 0.03$ | $\mathbf{49.20} \pm 0.97$ |
| $\omega = 0.05$ | $42.10 \pm 1.84$ |

30 epochs. Once the base model is obtained, we further fine-tune the last layer of the model for 10 epochs using Meta Sampler and Balanced Softmax (Ren et al., 2020). We use the SGD optimizer with a learning rate of 0.0028, momentum of 0.9 and weight decay 0.0005 and batch-size of 512. During generation, we apply dropout on image-embeddings with 0.5 probability. Furthermore, we apply 30 steps of DDIM with clip-guidance scale of 3 and feedback guidance scale of 0.03.

