# OpenReview forum: "Feedback-guided Data Synthesis for Imbalanced Classification"
_TMLR — Accepted by TMLR_

### Review · Reviewer_RUw5 · 2024-04-07

**Summary Of Contributions:**

The submission presents a novel framework that addresses the challenges of imbalanced classification in machine learning by augmenting static datasets with synthetic samples. The key contributions and new knowledge introduced are as follows:

1. **Feedback-guided Data Synthesis**: The authors propose a method that leverages one-shot feedback from a classifier to guide the sampling process of a generative model. This approach aims to synthesize samples that are useful for improving the classifier's performance, particularly for underrepresented classes.

2. **Framework Effectiveness**: The framework is designed to create synthetic samples that are close to the support of the real data distribution and sufficiently diverse. The authors validate the effectiveness of their approach on long-tailed datasets such as ImageNet-LT and Places-LT, as well as a group-imbalanced dataset NICO++.

3. **Feedback Criteria**: The authors explore three feedback criteria for sample generation - classifier's loss, prediction entropy, and hardness score. These criteria help in identifying and generating samples that are informative and challenging for the classifier.

4. **Dual Conditioning for Realism and Diversity**: To ensure that the synthetic samples align with the real data distribution, the authors use dual conditioning with both text prompts and real image embeddings. This technique helps to mitigate issues such as homonym ambiguity, text misinterpretation, and stylistic bias.

5. **Enhancing Diversity through Dropout**: To increase the diversity of synthetic samples, the authors propose applying random dropout on the image embedding. This method introduces variability in the generation process, leading to a more diverse set of samples that better represent the real-world data distribution.

**Audience:**

Yes

**Claims And Evidence:**

Yes

**Requested Changes:**

Proposed Adjustments:

1. **Classifier Guidance (Critical)**:
   - The submission should investigate and address the instances where classifier guidance does not yield competitive improvements.
   - It is essential to ensure that the classifier guidance is robust and consistently beneficial across various scenarios and datasets.

2. **Benchmarking Against Established Methods (Critical)**:
   - The authors should compare their guidance strategy with established methods, such as those leveraging pre-trained CLIP models, to demonstrate the superiority or unique benefits of their approach.

**Strengths And Weaknesses:**

Strong Aspects:

1. **Innovative Approach**: The submission introduces a novel framework that leverages feedback from classifiers to guide the synthesis of synthetic data, which is a creative approach to addressing the issue of imbalanced classification.

2. **Extensive Experiments and Ablation Study**: The authors conduct comprehensive experiments and an ablation study to demonstrate the impact of each component of their framework, providing strong evidence of its efficacy.

3. **Addressing Specific Challenges**: The submission tackles specific challenges in synthetic data generation, such as homonym ambiguity, text misinterpretation, and stylistic bias, with innovative solutions.

Weaknesses:

1. **The Necessity of Classifier Guidance**: A pivotal aspect of the methodology is the feedback from classifiers to guide the generation. However, this occasionally appears to be ineffective. Notably, the performance of LDM-FG, as outlined in Tables 2 and 3, does not present a competitive improvement. Particularly in Table 3, where the best result of LDM-FG is worse than that of vanilla LDM in terms of average accuracy. This observation raises questions about the utility of implementing such a complex procedure.

2. **Evaluation of Guidance Strategies**: The concept of employing auxiliary information to steer the generation of synthetic data is not unprecedented. For instance, previous work [1] has leveraged the capabilities of the pre-trained CLIP model to direct image generation tasks for classification of small-scale datasets. It prompts an inquiry into whether the approach of utilizing a classifier for guidance could outperform such established methods.

3. **Comparative Analysis with Alternative Approaches**: The research incorporates external data and models as part of its methodology, yet the resulting performance does not markedly exceed that don't utilize external resources, and is worse than methods using external data. Specifically, reference [2] demonstrates a significant improvement in results through the use of external data for model training. So when you use external information like LDM, the result so far is not promising.

[1]: Zhang, Yifan, et al. "Expanding small-scale datasets with guided imagination." Advances in Neural Information Processing Systems 36 (2024).

[2]: Tian, Changyao, et al. "Vl-ltr: Learning class-wise visual-linguistic representation for long-tailed visual recognition." European Conference on Computer Vision. Cham: Springer Nature Switzerland, 2022.

---

> ### Author Response · Authors · 2024-06-13
>
> Thank you for taking the time to review our work! We are happy that you found our work innovative with extensive experimental and ablation studies.
>
> ### **On NICO++ results**
>
> We believe that the reviewer might have made an oversight interpreting the results in Table 3.
> Firstly, in our results in Table 3, we report the average accuracy along with standard deviation. Note that we are outperforming other methods in terms of the worst group accuracy by a large margin. In terms of the **average accuracy**, comparing the standard deviations, we observe that our results compared to any other method are **statistically the same**, meaning that they are within the standard deviation of results reported for LDM.
>
> Secondly, in the NICO++ dataset, the task is a sub-population shift where we have 360 groups. For some of these groups we do not have any training examples. As a result, by definition these groups are minorities and have very little effect in terms of the overall average accuracy. Consequently, it is important to study the worst group accuracy as it provides a measurement on how well a method performs for the minority groups and how robust it is. The focus in the literature is often on the worst group accuracies as well. See [CHANGE] that studies a wide range of methods and approaches on the NICO++ dataset and often the focus is on worst group accuracy.
>
> [CHANGE] Change is Hard: A Closer Look at Subpopulation Shift Yang et al, 2023.
>
> ### **Comparison with alternative approaches**
>
> We would like to point out that our method is "data-driven" instead of "algorithm-driven", meaning we concentrate on enhancing the usefulness of the training data rather than relying on algorithmic changes. This difference is important because our approach can be combined with other learning algorithms to potentially achieve even better results. The papers pointed out by the reviewer are mostly algorithm driven which are complementary to our framework.
>
> To showcase that our method is complementary to  existing algorithmic/architecture approaches, we explored an additional approach proposed by the reviewer. Specifically, we follow the setup in [VL-LTR], and use the fine-tuned ViT-B16 model to get feedback and target ImageNet-LT task. We generate synthetic data using feedback guidance with entropy and create a training dataset such that the combination of synthetic + real data is 1.3 million examples.
>
> We then apply only 10 epochs of additional fine-tuning using the 1.3M data and observe that on classes few and medium, we improve the accuracies by 5 and 1 points respectively while maintaining the overall average performance. **Note that we also train the baseline model for an additional 10 epochs using ImageNet-LT data and observe slightly lower results than the one reported in the paper.
>
> This experiment validates that our framework is complementary to pre-existing architecture and algorithmic approaches and it is robust across different methods and architectures.  We have included these results in the main text as well. See Table 1 and Section 4.1.
>
> [VL-LTR] Tian, Changyao, et al. "Vl-ltr: Learning class-wise visual-linguistic representation for long-tailed visual recognition." European Conference on Computer Vision. Cham: Springer Nature Switzerland, 2022

---

### Review · Reviewer_sT9Z · 2024-05-20

**Summary Of Contributions:**

The paper proposes to generate samples for long-tailed classes to supplement the scarcity of samples by using feedback from the classifiers. Specifically, the feedback from the classifier is used as an input to the diffusion model to generate samples to supplement the dataset that the classifier is updated on. There are three types of feedback -- (1) classifier loss (cross entropy), (2) entropy of output class distributions and (3) hardness score (by Sehwag et al., 2022). By empirical validations with ImageNet-LT, Places-LT, NICO++, the proposed method improves the state of the arts by large margins in two datasets (ImageNet-LT, NICO++) but not with Places-LT.

**Audience:**

Yes

**Broader Impact Concerns:**

There is no ethical implication of the work.

**Claims And Evidence:**

Yes

**Requested Changes:**

**Suggestions**
- It may be worthwhile to try to learn the weighted combination of proposed three scores into a learned score (learning the mixing coefficients for the three scores to get a single score)
- It would be great to have a quantitative metric for measuring diversity of samples

**Minor suggestions**
- in Sec. 3.2, there is a mention about "a dual-conditioning technique" without a reference. Please add citation for it.
- In the paragraph in the section 1, starting with "Yet, the generative model literature...", there is 'prompt engineering' phrase. The first quotation mark should be ` in LaTeX instead of '.

**Strengths And Weaknesses:**

**Strengths**
- The method is simple but quite effective in terms of empirical gain.
- The paper is well written and easy to understand

**Weaknesses**
- The method does not have noticeable technical contribution. It is a combination of existing components or heuristic proposals (e.g., the three feedback criteria)
- Mixed empirical gains (not noticeable gain in Places-LT). But there is not much of discussion (although there is discussion subsections) for not compelling empirical gains by the proposed method.
- There is no quantitative measure for the diversity.  Without the metric, it is difficult to argue the generated samples are diverse.
- Lack of theoretical argument of why the proposed method works.

---

> ### Author Response · Authors · 2024-06-13
>
> Thanks for your time! We have incorporated your feedback and address your concerns below.
>
> ### **On the novelty and technical contribution**
>
> Although we think our work is novel and our experiments back this up, we want to highlight TMLR's guidelines: submissions shouldn't be assessed solely based on achieving new state-of-the-art results or being considered "novel enough". Instead, the focus is on whether the work offers useful insights for researchers in the field; we hope the reviewer finds that our work does indeed provide such value.
>
> To highlight our contributions, note that our work centers around introducing guidance through feedback: Although our method combines existing elements, the novel aspect is incorporating feedback from the classifier to guide the generative model. This feedback mechanism aims to enhance the usefulness and diversity of generated samples rather than merely using a generative model for data generation. The importance of ‘feedback guidance’ is particularly highlighted in long-tailed and group-imbalanced datasets. For example, as shown in our results where feedback-guided generation outperforms the Fill-Up baseline while generating only half as much data.
>
> ### **On Places-LT results**
>
> As shown in Table 2, using 'feedback-guided' generated samples for training leads to a notable improvement over the baseline techniques. Specifically, we would like to remind you that the goal in the existing baselines for imbalanced classification is to improve the accuracy of minority classes while also improving or maintaining the overall average accuracy. In our results for Places-LT, we improved the performance on “classes few” by 5%, compared to FILL-UP and 4% compared to the best algorithmic approaches (MiSLAS). This is a notable improvement on “classes few” while also improving the overall average accuracy.
>
> We would like to point out that our method is "data-driven" instead of "algorithm-driven", meaning we concentrate on enhancing the usefulness of the training data rather than relying on algorithmic changes.
>
> This difference is important because our approach can be combined with other learning algorithms (for example MiSLAS) to potentially achieve even better results.
>
> To showcase this point and also as suggested by reviewer RUw5, we have conducted additional experiments on ImageNet-LT that build upon previous work that leverage CLIP pre-training and use larger architectures such as the ViT-B model. We observed that our data approach combined with the algorithmic and architectural approach proposed in the [VL-LTR] paper, shows promising results. Specifically, we have shown that using feedback from a fine-tuned ViT-B model while using visual-linguistic long-tailed recognition framework and generating 1.3M synthetic examples, improves the accuracy on classes few and median by 5 and 1 points respectively while maintaining the overall average performance.
>
> In conclusion, while we have showcased the effectiveness of our framework, it is important to note that it can be combined with any existing and future algorithmic approaches for long-tailed recognition tasks.
>
> [VL-LTR] Tian, Changyao, et al. "Vl-ltr: Learning class-wise visual-linguistic representation for long-tailed visual recognition." European Conference on Computer Vision. Cham: Springer Nature Switzerland, 2022
>
> ### **Quantitative measure for the diversity**
>
> We believe the reviewer might have made an oversight inspecting the results of our work. Quantitative measurements of diversity are already included in table 4. The metrics of FID, Density, and Coverage as provided in Table 4 give a thorough quantitative evaluation of quality and diversity of the generations. FID measures the overall resemblance between real and generated samples' distributions, encompassing both quality and the diversity. Coverage measures diversity by determining how much the generated samples cover various modes of the real data distribution. See more discussion in Section 4.4.
>
> ### **On weighted combination of proposed three scores**
>
> Thank you for this suggestion. We have already conducted this experiment in our original manuscript and studied how the combination of different criteria affect the performance of the model (see section F.2 in the appendix).
> In this setup, for the three criteria studied in the paper (loss, hardness, entropy), we generate a dataset of size 1.3M by selecting each criteria with 0.33 probability for generating a sample. We tune the hyper-parameters of the classifier. You can find this result and details in Section F.2 of the paper. Overall, the entropy criterion is proven to be the best in all the datasets that we studied and outperforms using a combination of criteria.
>
>
> We have also included your additional minor suggestions. Thanks again for your feedback!

---

> > ### Comment · Reviewer_sT9Z · 2024-06-13
> > **Reply by the reviewer sT9Z**
> >
> > **On the novelty and technical contribution**
> >
> > The reviewer guideline suggests that the novelty should not be sole measurement for paper decision. First, I don't make a decision but give my review comments. Second, the authors contribution is not theoretically justified but a rather heuristically and intuitively built components. While I do appreciate the intuitively built method, these sort of methods are mostly justified their value by empirical validations. Therefore, it is common practice that the value or the insight of the work combining existing components would be validated by the empirical results, saying the combination is insightful and original with the support of empirical results.
> >
> > **On Places-LT results**, **Quantitative measure for the diversity** and **On weighted combination of proposed three scores**
> >
> > Thank you for the response. For the diversity, I do agree that there is no better measure than FID and coverage for now.

---

> > ### Comment · Reviewer_sT9Z · 2024-07-24
> > **Thank you for the detailed answers.**
> >
> > Thank you for the detailed answers. Except the novelty concerns, all of my concerns are resolved by the authors' response.

---

### Review · Reviewer_zK2a · 2024-05-29

**Summary Of Contributions:**

Some recent computer vision models use a generative model to augment static datasets with synthetic data. Current practice is limited by the lack of a feedback mechanism between the generative model and the final classifier. The authors use one-shot feedback from the classifier to the generative model to drive sampling useful synthetic examples. Long-tailed and group-imbalanced datasets are used as an evaluation, and often shows state of the art results.

**Audience:**

Yes

**Broader Impact Concerns:**

The method concerns generating samples in an imbalanced data regime (containing potentially protected classes), and using those synthetic generated examples to classify synthetic data. I would expect that such a model would have potential for serious intentional misuse (e.g. targetting minorities) or potential for serious unintentional misuse (e.g. exacerbating bias in imbalanced data with the false belief that this bias has been fixed). Sadly, neither of these broader impacts are even mentioned, let alone discussed. The only evaluations that are considered are classification accuracy and generative metrics (e.g. FID, density, coverage). I do not work in generative models so I am not sure, but I would have thought that broader impacts are a necessary consideration for such work.

**Claims And Evidence:**

Yes

**Requested Changes:**

Please address my questions above, particularly the Data Processing Inequality. Also please address the broader impact concerns below.

**Strengths And Weaknesses:**

Strengths:
- The paper presents a empirically-driven pipeline that seems intuitively matched to the task at hand. The reported experiments show state-of-the-art results.
- The paper cites a decent amount of related literature, and discusses them at a good amount of detail (although since this is not my area, I am not sure if they missed fundamental work).
- The text and illustrations are clear and effectively describe the method and results.
- Ablation is performed well. E.g. Table 1 and 2,3 and 4 shows that Entropy is probably the most important/useful conditioning signal (for the downstream task). This would not be obvious if the authors had chosen to omit the results using Loss and Hardness.

Weaknesses:
- What does the classical Data Processing Inequality say about trying to get more useful information out of the dataset that was used to train the generative model, than the dataset itself? If a generative model $f$ is trained on a dataset $X$, then the information contained within $f$ cannot be greater than the information in $X$. This would seem to imply that rather than using the generative model to make synthetic examples, we should just use the original data used to train the generative model instead. Why then does using a generative model for synthetic data work? I am not familiar with this area so it is possible that there are some fundamental works you should cite here to clarify the setup in light of the Data Processing Inequality. I tick "no" to claims and evidence primarily for this point - it could be an easy fix for the authors to respond to, perhaps a few sentences in the first paragraph of section 5.
- A key finding that is mentioned several times is that "the synthetic data must lie close to the support of the downstream task" and "be sufficiently diverse". I am not sure why this is surprising, or interesting. Isn't this rather an assumption of training an ML model? We classically assume data comes iid (and is therefore sufficiently diverse) from a fixed distribution (with a fixed support). Somehow this insight is already baked into the classical empirical risk minimisation setup. What is the new insight?
- No theory or conceptual analysis is provided, beyond the rough intuition understood from Figure 2.

---

> ### Author Response · Authors · 2024-06-13
>
> Thank you for your thoughtful feedback! We're happy you found our work intuitive and clear! Below we have addressed your questions.
>
>
> ### **On the Data Processing Inequality**
>
> There are a couple points that we would like to clarify.
>
> Firstly, in our experiments, we leverage a **pre-trained off-the-shelf** generator that has been trained on vast and diverse datasets (in an order of billions of examples). We use this generator to produce examples for an imbalance classification task. Our objective is to exploit this generator to create a customized training set that is beneficial for the specific task at hand rather than using all the billions of examples to train the classifier.
>
> Secondly, in many applications and scenarios, we often do not have access to these large scale datasets and even if we do, they often take up a large amount of memory. On the other hand, using an off-the-shelf pre-trained diffusion model is similar to having access to a compressed form of this data (in our setup, the diffusion model takes 0.00005 memory compared to the data it is trained on).
> Furthermore, in the original dataset, we may have many irrelevant samples to the task at hand, while feedback guided sampling from the diffusion model can be viewed as querying **useful** data from the large-scale corpse of data.
>
> In conclusion, in our setup, we assume easy access to large scale pre-trained diffusion models, from which we can get useful examples to improve the performance of our downstream classifier. As a result, it is desirable to generate **useful** data from the diffusion model and train the classifier on this smaller set, rather than training on billions of real examples that may be irrelevant to the task at hand.
> We have included more discussion around this in Section 6 of the main paper.
>
> ### **On the importance of "the synthetic data must lie close to the support of the downstream task"**
>
> The reviewer is correct that if the data is iid, then the above statement is obsolete. However, in our setup, we are not operating under the iid assumption. Our generator is a **pre-trained off-the-shelf** model, trained on a large corpse of data. So, the generator has the ability to produce a broad range of samples with different styles, many of which may not be relevant to the specific downstream task or they might have a different style compared to the real dataset. See Sec. 3.2 and Figure 4 where using only the text label results in examples that are not close to the real data distribution and we observe a distribution shift between real and synthetic examples.
>
> This is why we need additional steps in the generation pipeline to ensure that the generated samples lie close to the support of the downstream task. Note that this distribution shift is recognized in the literature. Authors in [FILL_UP, CVSG, CGEN] have also observed a similar issue and addressed the misalignment between synthetic and real examples using textual inversion or some form of contextualization. In our work we show that simply conditioning the generations on real examples (see more discussion in Sec. 3.2) leads to improved results.
>
> [FILL_UP] Fill-Up: Balancing Long-Tailed Data with Generative Models, Shin et al 2023
>
> [CVGS] Improving Geo-diversity of Generated Images with Contextualized Vendi Score Guidance, Hemmat et al 2024
>
> [CGEN] Dataset Interfaces: Diagnosing Model Failures Using Controllable Counterfactual Generation, Vendrow et al 2023
>
> ### **On Broader Impact Concerns**
>
> Thanks for your feedback. We understand the need to consider broader impacts, particularly when dealing with generative models and data imbalance scenarios and we have now included a broader impact section in the revised manuscript.
>
> We would like to highlight that in all our experiments we have shown improvements compared to previous baselines with respect to minority examples.  For all the experiments we have reported worst-group or minority class accuracies along with overall average accuracies. By focusing on these metrics, we directly target the reduction of bias and enhancing fairness in classification tasks. We have also included measures of diversity and coverage to assess how well the generated samples represent the targeted distribution. This is crucial to ensure that the synthetic data enhances the representation of minority classes rather than reinforcing existing imbalances. As mentioned, our evaluation metrics include worst-group accuracy, which directly measures the performance on the most challenging subsets of data, often corresponding to minority groups.
>
> Beyond the classification task, we agree that generative models have associated risks. In the revised version, we have added a separate section on “Broader Impacts” that discusses our work’s motivations and its broader impact.

---

> > ### Comment · Reviewer_zK2a · 2024-06-13
> >
> > Thanks very much for your response and updated manuscript.
> >
> > Regarding the data processing inequality, your response makes sense and it is clear that using a model that is pre-trained on a huge dataset has advantages over trying to extract information directly out of a huge dataset. The extra paragraph in section 6 is useful, and adequately addresses my concern.
> >
> > Thanks also for adding the comment on broader impact concerns.
> >
> > I've updated my review (ticking "Yes" and "Yes"), and will recommend acceptance.

---

### Author Response · Authors · 2024-06-13
**To all the reviewers**

We thank all the reviewers for their valuable feedback! We are happy that the reviewers recognized our method’s simplicity and effectiveness and appreciated the results and the ablation analysis. We have tried to address the questions asked by the reviewers below and directly made the requested changes in the revised draft (marked in blue). Furthermore, we have done an additional experiment on Image-Net-LT on a ViT-B model, trained with a visual-linguistic long-tailed recognition. See results in Table 1 and discussions in Sec. 4.1.

---

### Decision · Action_Editor_xA7M · 2024-07-29

**Recommendation:** Accept as is

**Comment:**

All reviewers agreed that the approach is simple and achieves good results at least in some important cases such as worst class accuracy and on some datasets made overall improvements on the performance. After author feedback, all reviewers converged to leaning acceptance of the paper, believing that although the method is simple the results are convincing enough to be published.

**Audience:**

Yes this is relevant for imbalanced classification.

**Claims And Evidence:**

This paper proposed to use classifier guidance to guide image synthesis as a simple way to improve synthesizing images for improving imbalanced classification. The claims are supported by experiments on several datasets.